# *HOPX* hypermethylation promotes metastasis via activating *SNAIL* transcription in nasopharyngeal carcinoma

Xianyue Ren[1,2,*], Xiaojing Yang[1,*], Bin Cheng[2,*], Xiaozhong Chen[3,*], Tianpeng Zhang[4], Qingmei He[1], Bin Li[3], Yingqin Li[1], Xinran Tang[1], Xin Wen[1], Qian Zhong[1], Tiebang Kang[1], Musheng Zeng[1], Na Liu[1] & Jun Ma[1]

Nasopharyngeal carcinoma (NPC) is characterized by a high rate of local invasion and early distant metastasis. Increasing evidence indicates that epigenetic abnormalities play important roles in NPC development. However, the epigenetic mechanisms underlying NPC metastasis remain unclear. Here we investigate aberrantly methylated transcription factors in NPC tissues, and we identify the *HOP* homeobox *HOPX* as the most significantly hypermethylated gene. Consistently, we find that HOXP expression is downregulated in NPC tissues and NPC cell lines. Restoring HOPX expression suppresses metastasis and enhances chemosensitivity of NPC cells. These effects are mediated by HOPX-mediated epigenetic silencing of *SNAIL* transcription through the enhancement of histone H3K9 deacetylation in the *SNAIL* promoter. Moreover, we find that patients with high methylation levels of *HOPX* exhibit poor clinical outcomes in both the training and validation cohorts. In summary, *HOPX* acts as a tumour suppressor via the epigenetic regulation of *SNAIL* transcription, which provides a novel prognostic biomarker for NPC metastasis and therapeutic target for NPC treatment.

[1] Sun Yat-sen University Cancer Center, State Key Laboratory of Oncology in South China, Collaborative Innovation Center of Cancer Medicine, 651 Dongfeng Road East, Guangzhou, Guangdong 510060, China. [2] Guangdong Provincial Key Laboratory of Stomatology, Guanghua School of Stomatology, Sun Yat-sen University, 56 Lingyuan Road west, Guangzhou, Guangdong 510055, China. [3] Department of Radiation Oncology, Zhejiang Cancer Hospital, 38 Guangji Road, Hangzhou, Zhejiang 310022, China. [4] Key Laboratory of Gene Engineering of the Ministry of Education, State Key Laboratory of Biocontrol, School of Life Science, Sun Yat-sen University, 132 Waihuan Road East, Guangzhou, Guangdong 510006, China. * These authors contributed equally to this work. Correspondence and requests for materials should be addressed to N.L. (email: liun1@sysucc.org.cn) or to J.M. (email: majun2@mail.sysu.edu.cn).

Nasopharyngeal carcinoma, which arises from the naso-pharynx epithelium, has a particularly high prevalence in South China[1]. To date, clinical therapeutic decisions regarding NPC patients are principally based on the tumour-node-metastasis (TNM) staging system. NPC patients with early stage should receive radiotherapy, while those with advanced disease should be treated with chemoradiotherapy[2]. Although NPC patients with the same TNM stage receive similar treatment regimens, approximately 30% of them eventually develop distant metastases, and the treatment outcomes of metastatic patients remain frustrating[3,4]. Epigenetic alterations are considered a hallmark of cancers because of their contribution to cancer initiation and progression[5,6]. The dynamic nature of DNA methylation makes it reversible and represents a promising prognostic biomarker and therapeutic target for cancers[7,8]. However, the aberrant methylation events of NPC metastasis remain unclear, and there is no effective forecasting model to accurately select patients with a high risk of metastasis.

Cancer is characterized by a subset of aberrantly expressed genes. Gene expression is regulated by many *cis*-regulatory elements, among which, transcription factors (TFs) play a leading role in initiation[9,10]. In contrast to the numerous signalling pathway genes, a limited list of dysregulated TFs in multiple cancers has been suggested as appropriate targets for the development of anticancer drugs[11]. The *HOP* homeobox (*HOPX*) was initially identified as a critical transcription factor for the modulation of cardiogenesis and development[12,13]. *HOPX* has been reported to play an essential role in the differentiation of various cells, such as lung alveolar cells[14], keratinocytes[15,16], T cells[17] and trophoblasts[18]. *HOPX* expression is epigenetically silenced in several malignant tissues, and the gene has been characterized as a tumour suppressor in lung cancer[14,19], gastric cancer[20] and colorectal cancer[21]. However, little is known regarding the mechanism of *HOPX* in cancer or its correlation with NPC.

Epithelial-to-mesenchymal transition (EMT) is a complex phenomenon in the biological processes that converts polarized, immotile epithelial cells to motile mesenchymal phenotypes. This conversion is believed to be triggered by EMT-TFs associated extracellular signals, including *SNAIL*, *SLUG*, *ZEB1*, *ZEB2*, *TWIST1* and *FOXC2*, and is thought to be a crucial process that enables cells to gain mobility and invasiveness for metastasis and acquire resistance to chemotherapy[22–24]. *SNAIL* is among the first TFs discovered to drive EMT via the direct inhibition of *ECADHERIN* transcription by binding to the E-boxes motif[25]. Given the critical roles of *SNAIL* in cell biological process, it becomes problematic when aberrantly activated in cancer. This issue has been supported by the finding that tumour cells with *SNAIL* overexpression possess metastatic and chemoresistance properties[26–29]. However, the mechanism of aberrantly expressed *SNAIL* remains elusive.

In the present study, we analysed the genome-wide methylation microarray data to identify potential NPC-specific TFs and determined that the methylation level of *HOPX* is the most significantly different TF in NPC tissues. *HOPX* suppresses metastasis and enhances chemosensitivity via the epigenetic inhibition of *SNAIL* transcription. Patients with *HOPX* high methylation exhibit poor clinical outcomes. In summary, our findings demonstrate a new *HOPX* regulation mechanism governing NPC progression, which may provide a novel prognostic biomarker and therapeutic target for NPC.

## Results

**The *HOPX* promoter is hypermethylated in NPC**. To identify potential NPC-specific TFs, we focused on the aberrantly methylated promoter regions of all human TFs between NPC ($n = 24$) and normal nasopharyngeal epithelial tissues ($n = 24$) in our previous genome-wide methylation microarrays[30]. Following the analysis of the microarray data, 1,984 CpG sites in the promoter regions of 344 TFs that differed significantly were found. After hierarchical clustering, the top 10 differentially methylated TFs were identified, among which *HOPX* (cg21899596) was the most significantly altered TF in NPC tissues (Fig. 1a and Supplementary Fig. 1a).

To confirm whether *HOPX* (cg21899596) was commonly hypermethylated, bisulfite pyrosequencing analysis was performed to examine the methylation levels in the other NPC ($n = 8$) and normal tissues ($n = 8$). The CpG islands and the selected region for bisulfite pyrosequencing in *HOPX* promoter region are shown in Fig. 1b. The methylation levels of *HOPX* (cg21899596) in NPC tissues were significantly increased compared with normal tissues (Fig. 1c,d). Similarly, *HOPX* (cg21899596) methylation levels in the NPC cell lines (SUNE1, CNE1, CNE2, HNE1 and HONE1) were substantially increased compared with the human immortalized normal nasopharyngeal epithelial cell (NPEC) lines (NP69, N2-Tert and N2-Bmi1; Fig. 1e and Supplementary Fig. 1b), indicating that *HOPX* is hypermethylated in NPC.

**HOPX downregulation is due to its promoter hypermethylation**. We subsequently investigated the association between *HOPX* expression and its promoter methylation status in NPC. First, we examined *HOPX* expression in fresh-frozen normal tissues and NPC tissues. Real-time RT–PCR (PCR with reverse transcription) and immunohistochemistry (IHC) results demonstrated that both the messenger RNA (mRNA; Fig. 2a) and protein (Fig. 2b,c) levels of *HOPX* were significantly downregulated in the NPC tissues. Notably, the HOPX protein levels were substantially lower in the NPC tissues with regional lymph node and distant metastasis than in the tissues without metastasis (Fig. 2b,c). In addition, we examined the *HOPX* expression in the NPEC and NPC cell lines via real-time RT–PCR and western blotting assays. Similarly, both the *HOPX* mRNA (Fig. 2d) and protein (Fig. 2e) levels were significantly downregulated in the NPC cell lines. To determine whether the downregulation of *HOPX* resulted from its promoter hypermethylation, the demethylation drug 5-aza-2′-deoxycytidine (Decitabine, DAC) was used. Following DAC treatment, the *HOPX* methylation levels were substantially decreased (Fig. 2f and Supplementary Fig. 2a), while the *HOPX* mRNA levels were significantly increased (Fig. 2g) in NPC cells compared with the NPEC cells. Taken together, these findings suggest that *HOPX* is downregulated in NPC, especially in cases of metastasis. The downregulation of *HOPX* is associated with the hypermethylation of its promoter in NPC.

**HOPX suppresses NPC cell migration and invasion *in vitro***. CNE2, SUNE1 and HONE1 cells were stably transfected with the control vector or *HOPX* expression plasmid to investigate the role of *HOPX* in malignancy. To determine whether *HOPX* regulated the migration and invasion of NPC cells, wound healing and Transwell assays with or without Matrigel were used. The ectopic expression of HOPX significantly suppressed NPC cells migration (Fig. 3a,b and Supplementary Fig. 3a–c) and invasion (Fig. 3c). In contrast, silencing HOPX using two different *HOPX*-siRNAs (small interfering RNAs) increased the migration (Fig. 3d–f) and invasion (Fig. 3g,h) of NPEC cells, as well as CNE2 and SUNE1 cells with stable HOPX overexpression (referred to as CH and SH cells, respectively; Supplementary Fig. 4a–c). We also investigated the effects of *HOPX* on cell growth. MTT and colony-formation assays demonstrated that the overexpression of HOPX in NPC

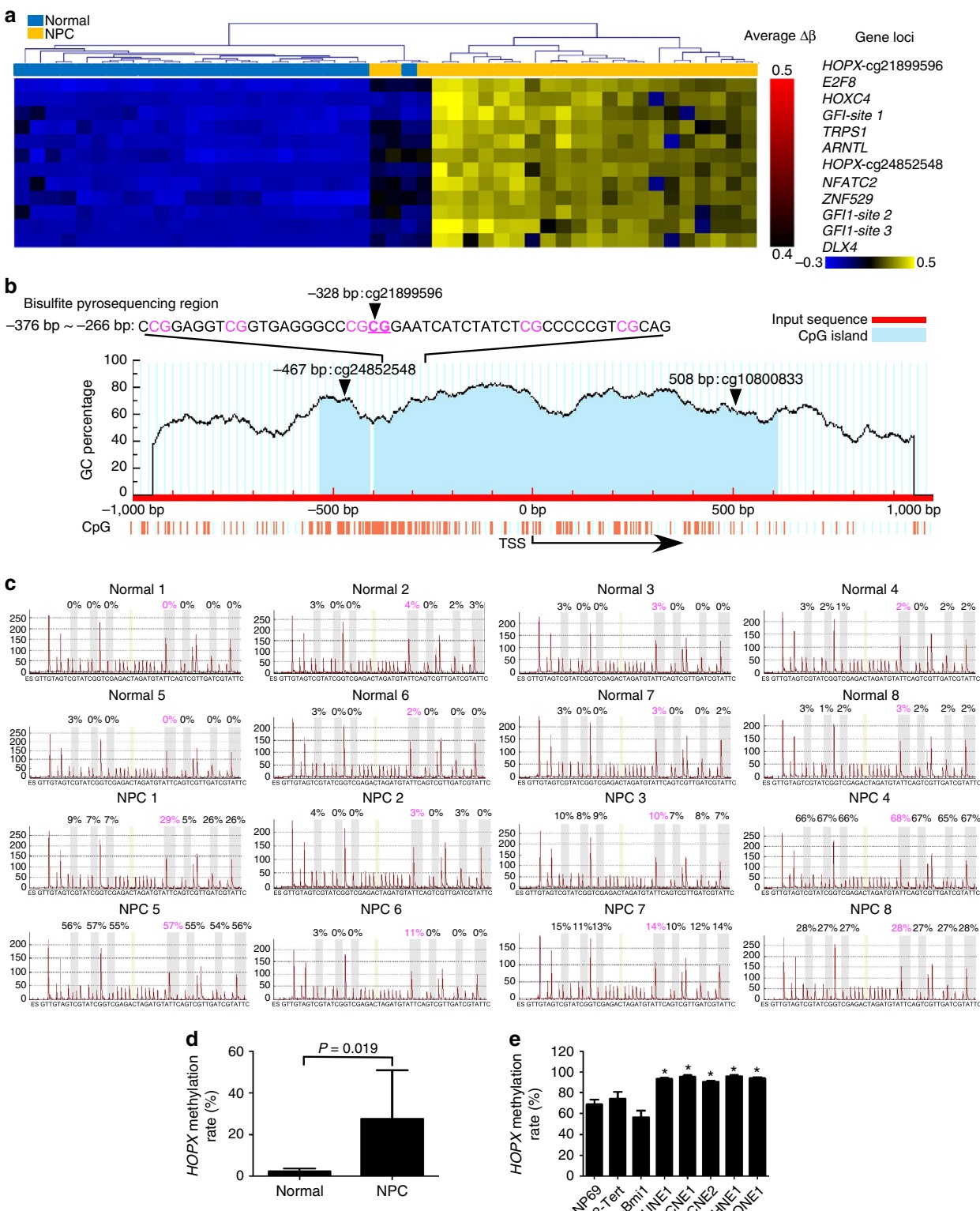

**Figure 1 | *HOPX* is hypermethylated in NPC.** (**a**) Heatmap clustering of the top 10 differentially methylated TFs CpG sites in normal nasopharyngeal epithelial tissues (normal, $n = 24$) and NPC tissues ($n = 24$). Columns: individual samples; rows: CpG sites; blue: low methylation; yellow: high methylation. (**b**) Schematic of the CpG islands and bisulfite pyrosequencing region in the *HOPX* promoter. Input sequence: red region; CpG islands: blue region; TSS: transcription start site; cg21899596, cg24852548 and cg10800833: the CG sites of *HOPX* identified in our previous genome-wide methylation microarrays; magenta words: CG sites for bisulfite pyrosequencing; bold magenta words: the most significantly altered CG site in *HOPX*. (**c**,**d**) Bisulfite pyrosequencing analysis of the *HOPX* promoter region (**c**) and the average methylation levels (**d**) in normal ($n = 8$) and NPC ($n = 8$) tissues. Magenta words: CG site of cg21899596. Mean ± s.d.; Student's *t*-tests. (**e**) The methylation levels of the *HOPX* promoter region, as determined by bisulfite pyrosequencing analysis, in NPEC (NP69, N2-Tert and N2-Bmi1) and NPC (SUNE1, CNE1, CNE2, HNE1 and HONE1) cell lines. Mean ± s.e.m.; *$P < 0.05$ compared with NP69; Student's *t*-tests. These data are representative of three independent experiments.

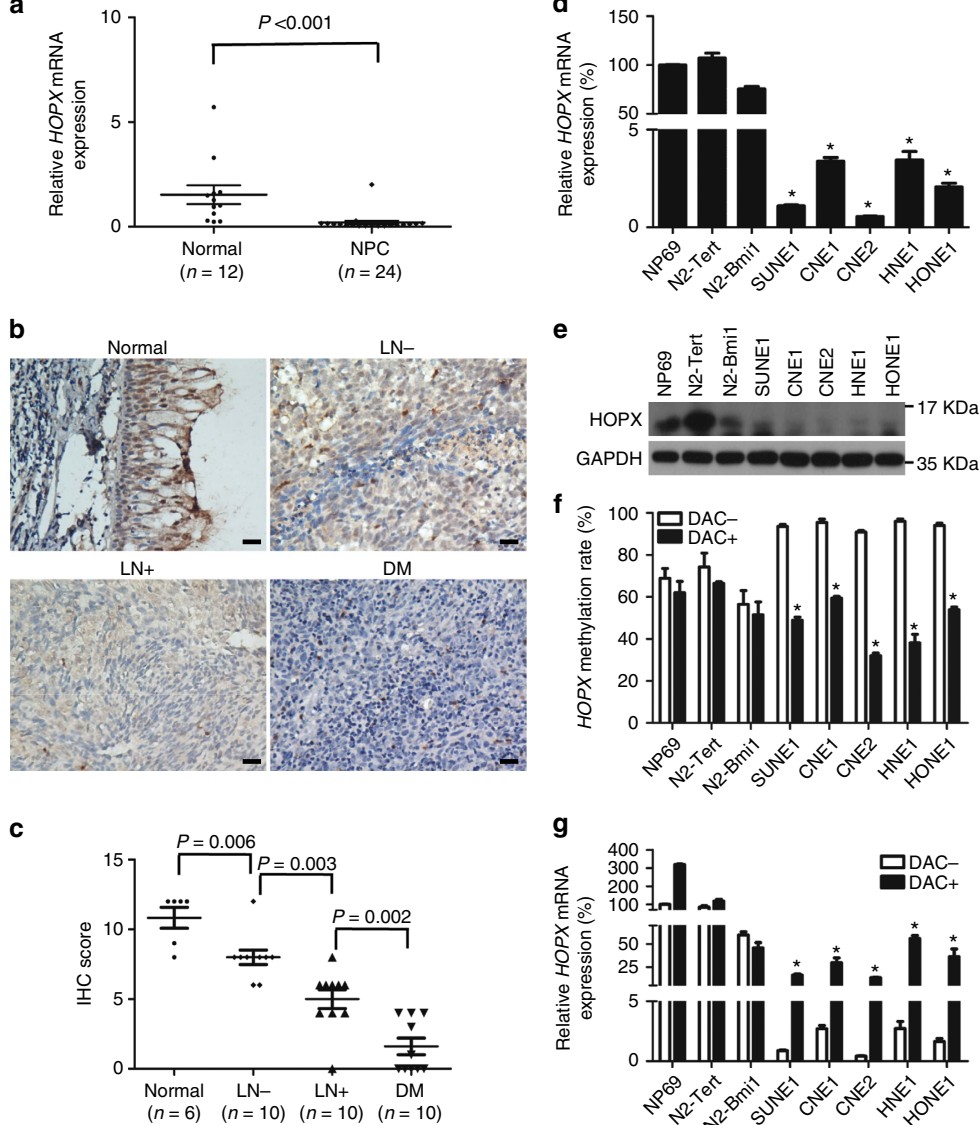

**Figure 2 | Reduced *HOPX* expression is associated with its promoter hypermethylation in NPC.** (**a**) Real-time RT–PCR analysis of relative *HOPX* mRNA expression in normal ($n = 12$) and NPC ($n = 24$) tissues. Mean ± s.d.; Student's *t*-tests. (**b,c**) Immunohistochemical staining (× 200) (**b**) and statistical analysis (**c**) of HOPX in Normal tissues ($n = 6$), primary NPC tissues without (LN − , $n = 10$) or with (LN + , $n = 10$) regional lymph node metastasis and primary NPC tissues with distant metastasis (DM, $n = 10$). Scale bar: 100 μm; mean ± s.d.; Student's *t*-tests. (**d,e**) The mRNA (**d**) and protein (**e**) levels of *HOPX* were measured via real-time RT–PCR and western blotting in NPEC and NPC cell lines. Mean ± s.d.; *$P < 0.01$ compared with NP69; Student's *t*-tests. (**f,g**) *HOPX* methylation levels measured via bisulfite pyrosequencing analysis (**f**) and relative *HOPX* mRNA levels measured via real-time RT–PCR analysis (**g**) with (DAC + ) or without (DAC − ) DAC treatment in NPEC and NPC cell lines. Mean ± s.e.m.; *$P < 0.01$ compared with DAC − ; Student's *t*-tests. These data are representative of three independent experiments.

cells or its silencing in NPEC cells had minimal effects on cell viability and colonization (Supplementary Fig. 5a,b; $P > 0.05$). Collectively, these findings illustrate that *HOPX* suppresses NPC cell migration and invasion and has a limited effect on cell growth *in vitro*.

**HOPX represses EMT in NPC cells *in vitro*.** Notably, when the expression of HOPX was knocked down in N2-Tert, NP69 and SH cells, we found that the morphology of some cells transitioned from an epithelial-like form to a spindle-shaped or elongated, mesenchymal form (Fig. 4a and Supplementary Fig. 6a), which indicated that HOPX functioned to maintain the epithelial status of NPEC and NPC cells. Therefore, we speculated that EMT might be involved in the suppression effect of HOPX on NPC cell

invasiveness. To explore the possible downstream molecular events of HOPX, a tumour metastasis PCR array was used. Among the top 10 differentially expressed genes, *ECADHERIN* was significantly upregulated in SUNE1 cells with HOPX overexpression (Supplementary Table 1 and Supplementary Fig. 6b), which indicated that HOPX might repress the EMT of NPC cells.

EMT has been reported to allow cells to acquire migratory and invasive behaviours during many biological processes, such as embryonic development, fibrosis and cancer metastasis[31]. Next, we performed immunofluorescent staining and western blotting assays to examine the effect of HOPX on the expression of epithelial markers (ECADHERIN and a-CATENIN) and mesenchymal markers (VIMENTIN and FIBRONECTIN). Laser scanning confocal microscope showed that the N2-Tert

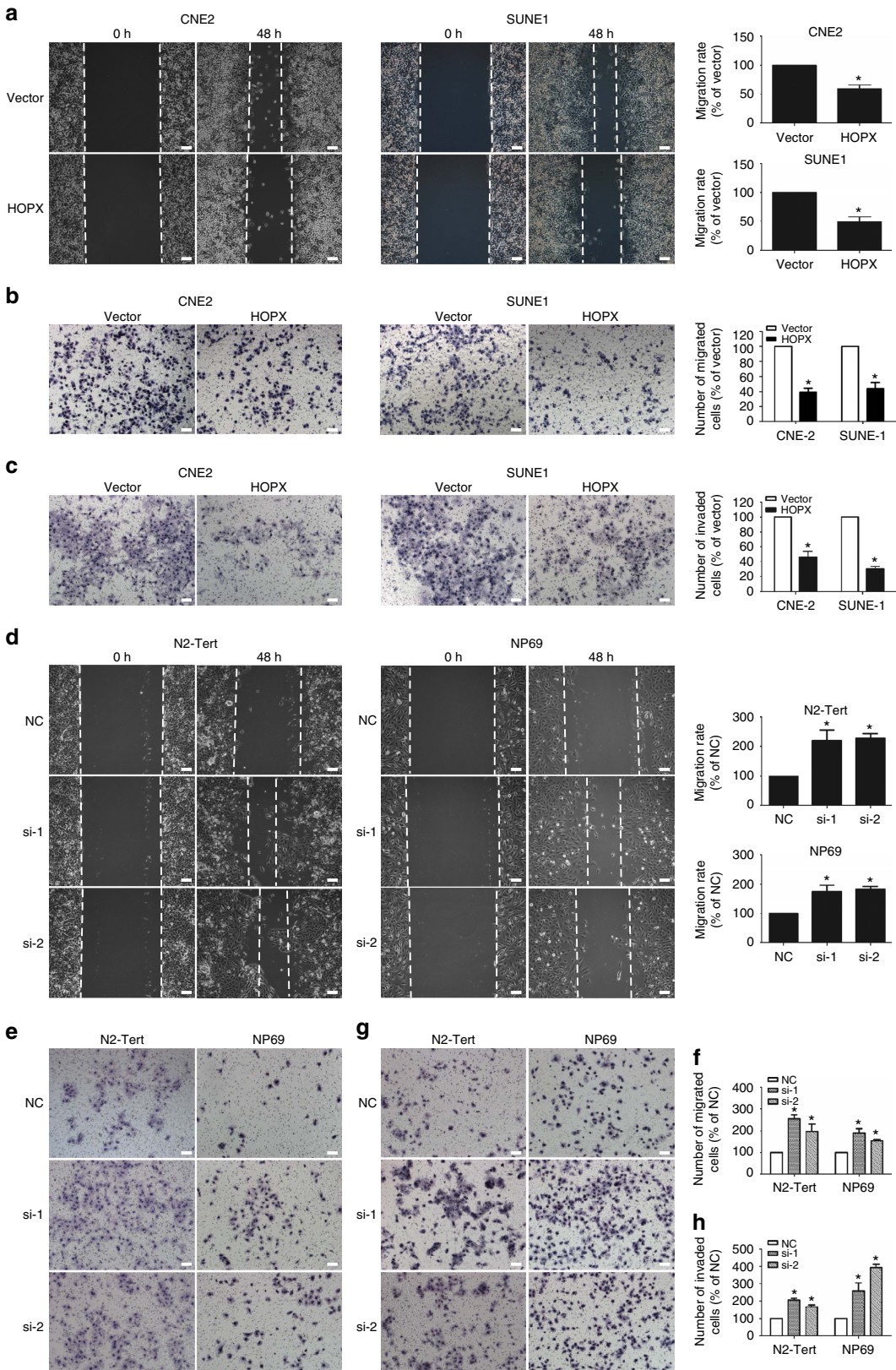

**Figure 3 | HOPX suppresses NPC cell migration and invasion in vitro.** (a–c) In CNE2 and SUNE1 cells that stably overexpressed the vector or HOPX, migration abilities were measured using a wound healing assay (×200) (a) and Transwell assay (×200) without Matrigel (b). Invasion abilities were measured using a Transwell assay with Matrigel (×200) (c). Scale bar, 100 μm; mean ± s.d.; *P < 0.01 compared with vector; Student's t-tests. (d–h) In N2-Tert and NP69 cells transiently expressing control NC or HOPX-siRNAs (si-1 and si-2), migration abilities were measured using a wound healing assay (×200) (d) and Transwell assay (×200) without Matrigel (e,f). Invasion abilities were measured using a Transwell assay with Matrigel (×200; g,h). Scale bar, 100 μm; mean ± s.d.; *P < 0.01 compared with NC; Student's t-tests. These data are representative of three independent experiments.

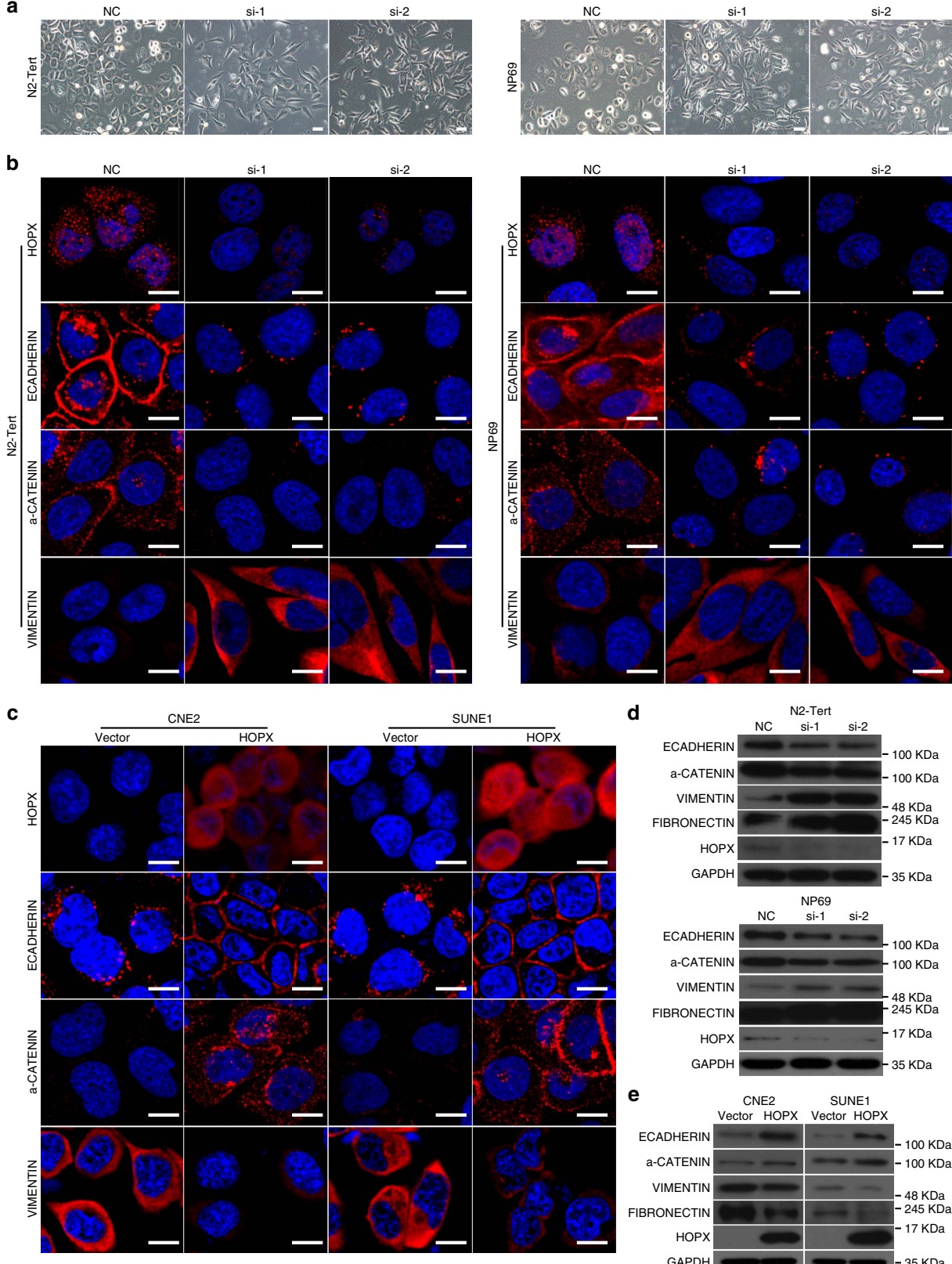

**Figure 4 | HOPX partially suppresses EMT in NPC cells.** (**a**) Phase contrast images (×200) of N2-Tert and NP69 cells expressing control NC or *HOPX*-siRNAs (si-1 and si-2). Scale bar, 100 μm. (**b,c**) Immunofluorescence images (×600) for HOPX, ECADHERIN, α-CATENIN and VIMENTIN expression in N2-Tert and NP69 cells transiently expressing control NC or *HOPX*-siRNAs (si-1 and si-2) (**b**), and CNE2 and SUNE1 cells stably overexpressed the vector or HOPX (**c**). Scale bar, 100 μm. (**d,e**) Western blotting analysis of ECADHERIN, α-CADHERIN, VIMENTIN, FIBRONECTIN, HOPX and GAPDH expression in N2-Tert and NP69 cells transfected with the control NC or *HOPX*-siRNAs (**d**), and CNE2 and SUNE1 cells stably overexpressed the vector or HOPX (**e**). These data are representative of three independent experiments.

and NP69 cells with HOPX silenced displayed reduced epithelial markers and increased mesenchymal marker (Fig. 4b). In contrast, the epithelial markers were upregulated, and the mesenchymal marker was downregulated in CNE2 and SUNE1 cells with HOPX overexpression (Fig. 4c). These findings were validated by western blotting assays (Fig. 4d,e). Together, these results suggest that *HOPX* suppresses EMT in NPC cells.

**HOPX suppresses the transcription of *SNAIL* in NPC**. Considering the critical roles of EMT-TFs in the process of EMT and metastasis, to further determine the target genes of *HOPX*, we examined the levels of the known EMT-TF regulators (*SNAIL*, *SLUG*, *ZEB1*, *ZEB2*, *TWIST1* and *FOXC2*) after altering HOPX expression[31,32]. Among all EMT regulators, *SNAIL* was the only regulator that exhibited significant changes in all investigated cell lines (Supplementary Fig. 7a,b). Both real-time RT–PCR and western blotting assays indicated that *SNAIL* was repressed when HOPX was restored in NPC cells and was increased when HOPX was silenced in NPEC cells (Fig. 5a–d). We subsequently investigated whether HOPX regulated the transcriptional activity of *SNAIL*. Thus, a *SNAIL* promoter reporter plasmid was constructed and transfected into NPC cells. Ectopic HOPX expression significantly decreased the luciferase activity of pGL3-*SNAIL* (Fig. 5e).

HOPX has no DNA-binding domain and cannot bind directly to DNA; however, it can indirectly bind to the promoter of a target gene by interacting with other transcriptional regulators[33]. To further investigate the role of HOPX in *SNAIL* expression, a chromatin immunoprecipitation (ChIP)-PCR assay was performed to determine whether the *SNAIL* promoter region (from $-1$ bp to approximately $-1,000$ bp upstream of the exon) was occupied by HOPX. Interestingly, only one binding site ($-521$ bp to approximately $-680$ bp) was found to interact strongly with HOPX in CNE2 and SUNE1 cells with exogenous HOPX expression (CH and SH cells), which was also identified in NPEC cells with endogenous HOPX expression (N2-Tert and NP69 cells) (Supplementary Fig. 7c–e). These binding effects were quantified by ChIP real-time PCR in Fig. 5f. Therefore, our findings imply that HOPX suppresses the transcription of *SNAIL* in NPC.

**HOPX suppresses *SNAIL* transcription via HDAC2 recruitment**. To explore the mechanism of *SNAIL* transcription repression mediated by HOPX, immunoprecipitation plus mass spectrometry was performed in SUNE1 cells with stable HOPX overexpression. Numerous HOPX-interacting proteins were identified, among which histone deacetylases 2 (HDAC2) was the only member of the HDACs family that might be responsible for the transcriptional repression effects of HOPX (Supplementary Data 1). The co-immunoprecipitation (Co-IP) assay verified that HOPX could physically interact with HDAC2 in CH, SH, N2-Tert and NP69 cells (Fig. 5g and Supplementary Fig. 8a). Western blotting assay showed that HOPX had little effect on HDAC2 expression (Supplementary Fig. 8b). To investigate whether HOPX promotes the recruitment of HDAC2 to the *SNAIL* promoter and is responsible for its histone deacetylation, ChIP real-time PCR assays were conducted to determine the binding efficiencies of HDAC2 and H3K9Ac to the *SNAIL* promoter. Following the overexpression of HOPX in CNE2 and SUNE1 cells, the enrichment of the *SNAIL* promoter region by the anti-HDAC2 antibody was substantially increased (Fig. 5h), whereas its enrichment by the anti-H3K9Ac antibody was decreased (Fig. 5i). Thus, we considered whether trichostatin A (TSA, a HDAC inhibitor) could affect *SNAIL* expression in NPC cells with HOPX overexpression. Our results demonstrated that

the repression effect of HOPX on *SNAIL* expression was significantly attenuated in a dose-dependent manner by TSA (Fig. 5j). In addition, the invasion ability and EMT of NPC cells with stable HOPX overexpression could be dramatically enhanced following siHDAC2 treatment (Supplementary Fig. 8c,d). Thus, these findings indicate that HOPX inhibits *SNAIL* expression via the interaction with HDAC2 and the recruitment of histone deacetylase activity.

**HOPX suppresses SRF-dependent *SNAIL* transcription**. HOPX functions as a corepressor in regulating gene expression through interactions with other TFs[33,34]. By examining the promoter region of *SNAIL* bound to HOPX, we identified a serum response element (*SRE*) using JASPAR software (Fig. 6a), which was the DNA-binding motif of serum response factor (SRF). Furthermore, SRF was identified as a HOPX-interacting protein in the mass spectrometry assay (Supplementary Data 1). These findings raised the possibility that SRF, as a transcription factor, might recruit HOPX to the *SNAIL* promoter. To test this hypothesis, we first examined the effects of SRF on *SNAIL* and *ECADHERIN* expression. The overexpression of SRF in CNE2 and SUNE1 cells increased *SNAIL* mRNA levels and decreased *ECADHERIN* mRNA levels (Fig. 6b). A mutant of the *SNAIL* promoter reporter gene that silenced the *SRE* motif was then constructed. The luciferase activities of the wild-type *SNAIL* promoter reporter gene increased significantly following transfection with *SRF*, whereas the activities of the mutant reporter gene were not affected (Supplementary Fig. 9a), confirming that SRF regulates the transcriptional activity of *SNAIL* in NPC cells.

Next, we performed Co-IP using an anti-HOPX antibody to test the physical interaction between SRF and HOPX, and the results showed that HOPX physically interacted with SRF in CH, SH, N2-Tert and NP69 cells (Fig. 6c). The NPC cells with HOPX overexpression recruited more SRF (Supplementary Fig. 9b), and showed little difference on SRF expression compared with cells containing the vector (Supplementary Fig. 9c). To verify that HOPX regulates *SNAIL* transcription by indirectly binding with the *SRE* motif, we also assessed the luciferase activities of a mutant of the *SNAIL* promoter reporter gene. Following mutation of the *SRE* motif, HOPX did not efficiently affect *SNAIL* transcription (Supplementary Fig. 9d). ChIP real-time PCR assay using an anti-SRF antibody confirmed that HOPX restoration attenuated the interaction between SRF and the *SNAIL* promoter (Fig. 6d). Moreover, we examined the ability of HOPX to modify the SRF activation of *SNAIL* luciferase reporter constructs. The co-transfection of *HOPX* and *SRF* markedly inhibited SRF-mediated *SNAIL* transactivation, whereas the activities of the mutant reporter gene were not affected (Fig. 6e). Real-time RT–PCR demonstrated that transfection with *SRF* resulted in increased *SNAIL* mRNA expression, and this activation was partly impaired by co-transfection with *HOPX* (Fig. 6f and Supplementary Fig. 9e). In addition, restoring SRF expression in NPC cells with stable HOPX overexpression could significantly enhance cell invasion and EMT (Supplementary Fig. 9f,g).

To confirm whether HOPX epigenetically suppresses SRF-dependent *SNAIL* transcription by recruiting histone deacetylase activity, we assessed the effects of TSA on SRF-dependent *SNAIL* transcription in NPC cells with HOPX overexpression. As expected, the inhibition effects of HOPX on SRF-dependent *SNAIL* mRNA expression were substantially attenuated by TSA (Fig. 6g). Moreover, we also found the co-location of HOPX, HDAC2 and SRF in NPC cells with stable HOPX overexpression (Supplementary Fig. 10a). Collectively, these findings demonstrate that HOPX inhibits NPC EMT and

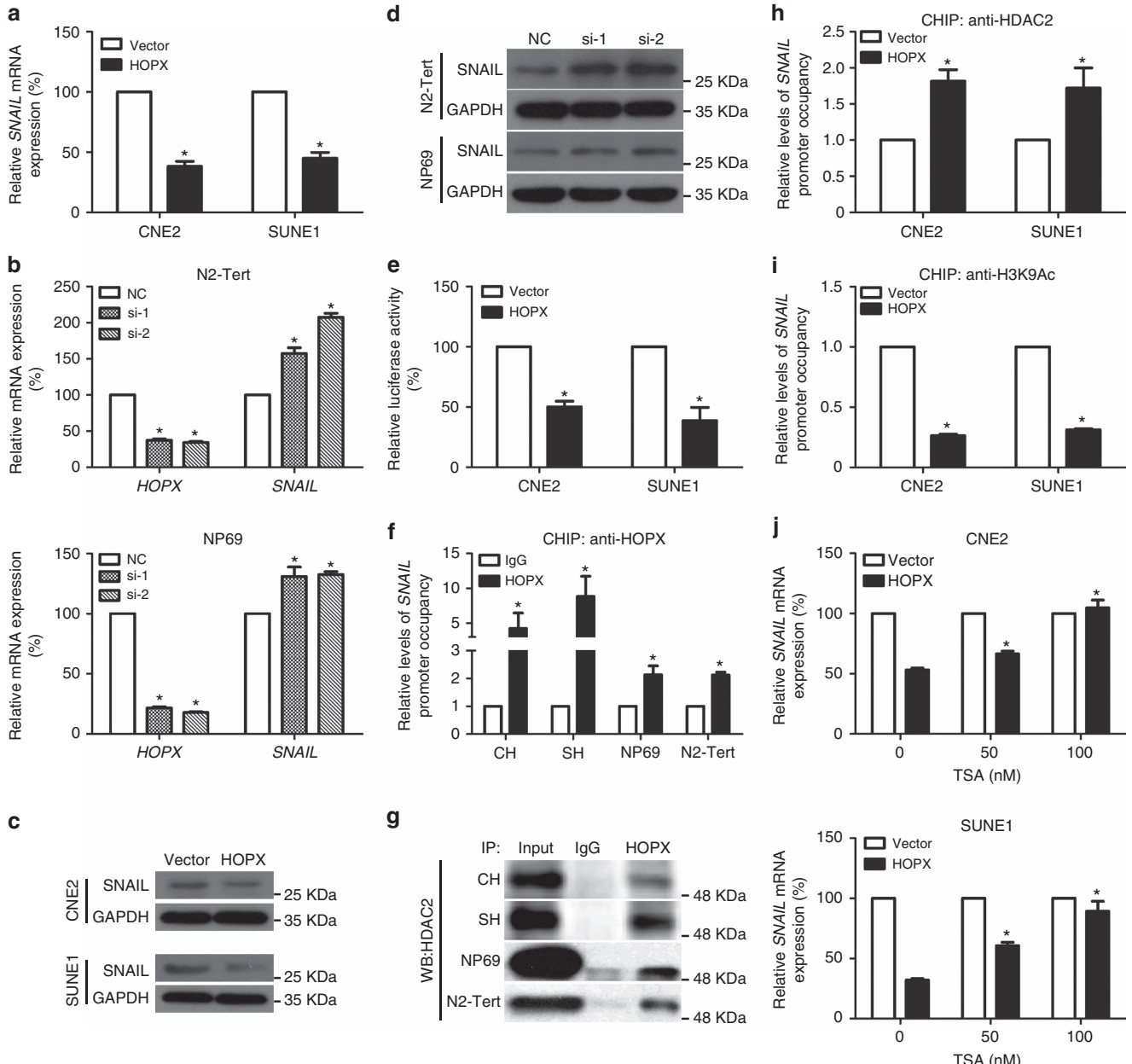

**Figure 5 | HOPX recruits HDAC activity to the *SNAIL* promoter and epigenetically suppresses *SNAIL* transcription in NPC.** (**a–d**) Real-time RT–PCR (**a,b**) and western blotting (**c,d**) assays were used to examine the mRNA and protein levels of *SNAIL* in CNE2 and SUNE1 cells stably overexpressed the vector or HOPX, or N2-Tert and NP69 cells transfected with the control NC or *HOPX*-siRNAs (si-1 and si-2). Mean ± s.d.; *$P < 0.01$ compared with vector or NC; Student's *t*-tests. (**e**) CNE2 and SUNE1 cells with transfection of the pGL3-*SNAIL* promoter were co-transfected with the vector or *HOPX* and subjected to dual-luciferase reporter assays. Mean ± s.d.; *$P < 0.01$ compared with vector; Student's *t*-tests. (**f**) ChIP real-time PCR assays were conducted to assess the enrichment of HOPX in the *SNAIL* promoter region in CH, SH, NP69 and N2-Tert cells. Mean ± s.d.; *$P < 0.01$ compared with IgG; Student's *t*-tests. (**g**) Physical interactions between HOPX and HDAC2 were examined via a Co-IP assay in CH, SH, NP69 and N2-Tert cells; WB: western blotting assay. (**h,i**) ChIP real-time PCR assays were applied to measure the enrichment of HDAC2 (**h**) and H3K9Ac (**i**) in the *SNAIL* promoter of CNE2 and SUNE1 cells stably overexpressed the vector or HOPX. Mean ± s.d.; *$P < 0.01$ compared with vector; Student's *t*-tests. (**j**) Real-time RT–PCR was conducted to assess the relative *SNAIL* mRNA expression in CNE2 and SUNE1 cells stably overexpressed the vector or HOPX that were treated with TSA (0, 50 or 100 nM). Mean ± s.d.; *$P < 0.01$ compared with TSA (0 nM); Student's *t*-tests. CH and SH cells indicated CNE2 and SUNE1 cells with stable HOPX overexpression. These data are representative of three independent experiments.

invasiveness via the histone deacetylase-mediated transcriptional repression of SRF-dependent *SNAIL* transcription.

**SNAIL is a functional target of HOPX in NPC.** To determine whether HOPX-mediated SNAIL downregulation contributed to the inhibition of NPC cell migration and invasion, we restored SNAIL expression in NPC cells that stably overexpressed HOPX.

Co-transfection with *SNAIL* significantly abolished the inhibitory effects of HOPX on NPC cells migration (Fig. 7a,b) and invasion (Fig. 7c). In addition, the expression level of ECADHERIN induced by HOPX was substantially decreased, and VIMENTIN was increased following co-transfection with *SNAIL* (Fig. 7d). These findings illustrate that *SNAIL* is a functional target of HOPX in NPC cells.

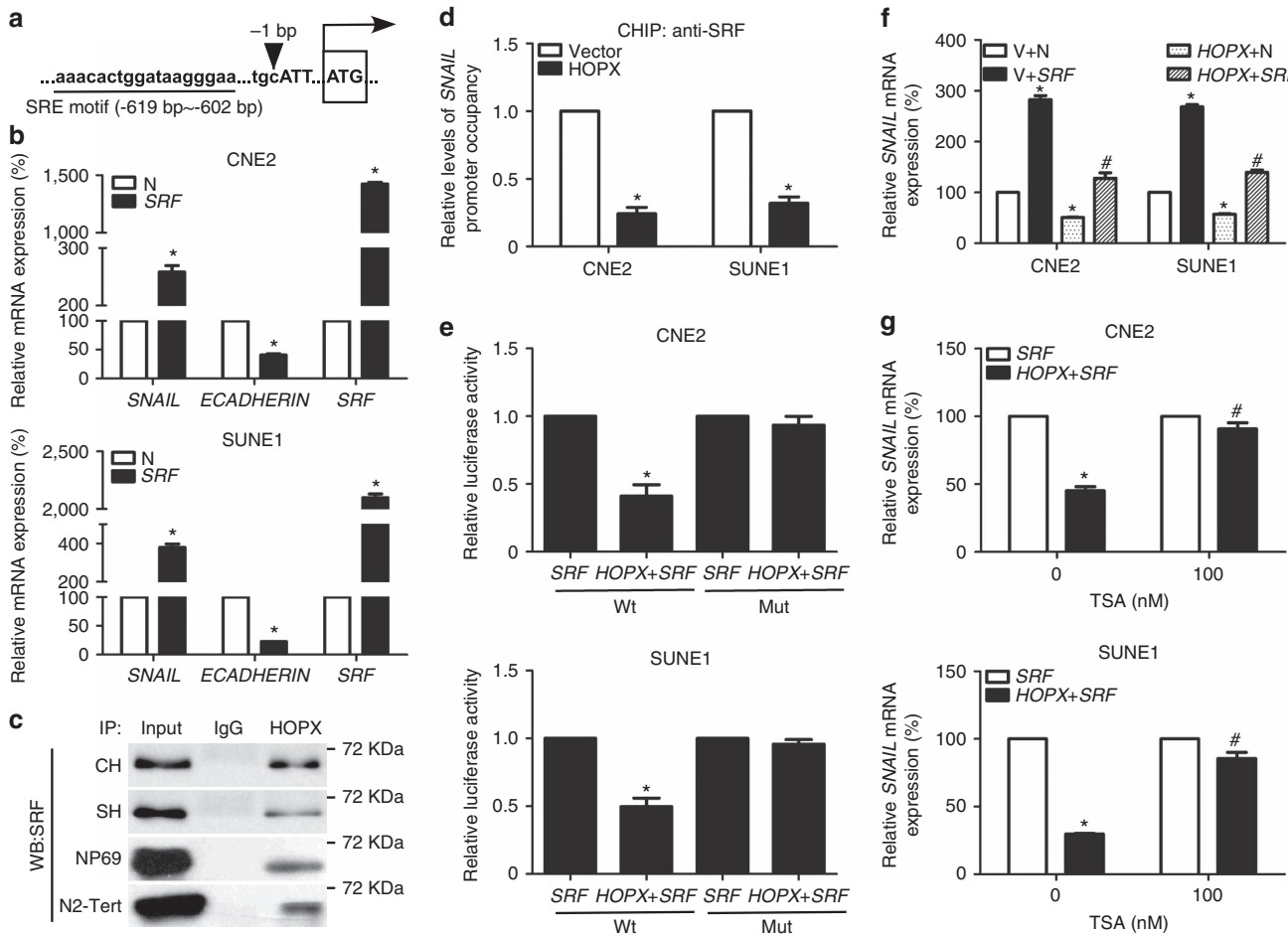

**Figure 6 | HOPX epigenetically inhibits SRF-mediated *SNAIL* transcription in NPC.** (**a**) Sketch map of the *SRE* motif within the *SNAIL* promoter predicted by JASPAR software. Upper case letters: exon. (**b**) Relative *SNAIL*, *ECADHERIN* and *SRF* mRNA expression levels were determined via real-time RT–PCR in CNE2 and SUNE1 cells stably overexpressed the control N or *SRF*. Mean ± s.d.; *$P < 0.01$ compared with N; Student's *t*-tests. (**c**) Co-IP assays were used to measure the interaction between HOPX and SRF in CH, SH, NP69 and N2-Tert cells. CH and SH cells indicated CNE2 and SUNE1 cells with stable HOPX overexpression; WB, western blotting assay. (**d**) ChIP real-time PCR assays were applied to measure the enrichment of SRF in the *SNAIL* promoter in CNE2 and SUNE1 cells stably overexpressed the vector or *HOPX*. Mean ± s.d.; *$P < 0.01$ compared with vector; Student's *t*-tests. (**e**) Wild-type and mutant *SNAIL*-luciferase reporters were constructed and co-transfected with *SRF* or/and *HOPX* in CNE2 and SUNE1 cells. Luciferase reporter assay was used to detect the luciferase activity of *SNAIL* promoter. Mean ± s.d.; *$P < 0.01$ compared with *SRF*; Student's *t*-tests. (**f**) CNE2 and SUNE1 cells were co-transfected with *HOPX* or/and *SRF*. V and N were used as empty vectors of *HOPX* and *SRF*, respectively. Relative *SNAIL* mRNA expression was measured via real-time RT–PCR. Mean ± s.d.; *$P < 0.01$ compared with V + N; #$P < 0.01$ compared with *HOPX* + N; Student's *t*-tests. (**g**) Real-time RT–PCR was applied to assess the relative *SNAIL* mRNA expression following transfection with *SRF* or/and *HOPX*. TSA (0 and 100 nM) was treated with CNE2 and SUNE1 cells. Mean ± s.d.; *$P < 0.01$ compared with *SRF*; #$P < 0.01$ compared with TSA (0 nM); Student's *t*-tests. These data are representative of three independent experiments.

**HOPX inhibits NPC aggressiveness *in vivo*.** To determine whether HOPX affected NPC cell invasion and lymph node metastasis *in vivo*, an inguinal lymph node metastasis model was constructed[35]. SUNE1 cells that stably overexpressed the vector or HOPX were constructed and injected into the foot pads of mice. After 6 weeks of growth, the primary foot pad tumours and inguinal lymph nodes were obtained ($n = 8$ per group, Fig. 8a). The volumes of the primary foot pad tumours in HOPX overexpression group had no significant difference compared with the vector group (Supplementary Fig. 11a; $P > 0.05$). Haematoxylin and eosin (H&E) staining of the primary tumours showed that tumours in the HOPX overexpression group exhibited sharp edges that expanded as spheroids, indicating a less aggressive phenotype with invasion towards the skin, muscle and lymphatic vessel than the vector group (Fig. 8b and Supplementary Fig. 11b). Moreover, the inguinal lymph nodes in the HOPX overexpression group had smaller volumes and fewer pan-cytokeratin-positive tumour cells than those in the vector group (Fig. 8c,d). In addition, the inguinal lymph nodes metastasis ratio was significantly lower in the HOPX overexpression group (Fig. 8e). Taken together, these findings imply that HOPX suppresses NPC cells invasion and lymph node metastasis *in vivo*.

In addition, a lung colonization model was used to determine whether HOPX restoration affected NPC tumour progression *in vivo*. SUNE1 cells that stably overexpressed the vector or HOPX were constructed and injected into the tail veins of nude mice. Six weeks later, the mice were killed, and the lungs were excised. Compared with the control group, the mice in the HOPX overexpression group had fewer tumour nodes on their lung surfaces (Fig. 8f). Moreover, H&E staining confirmed that the HOPX overexpression group had smaller and fewer microscopic

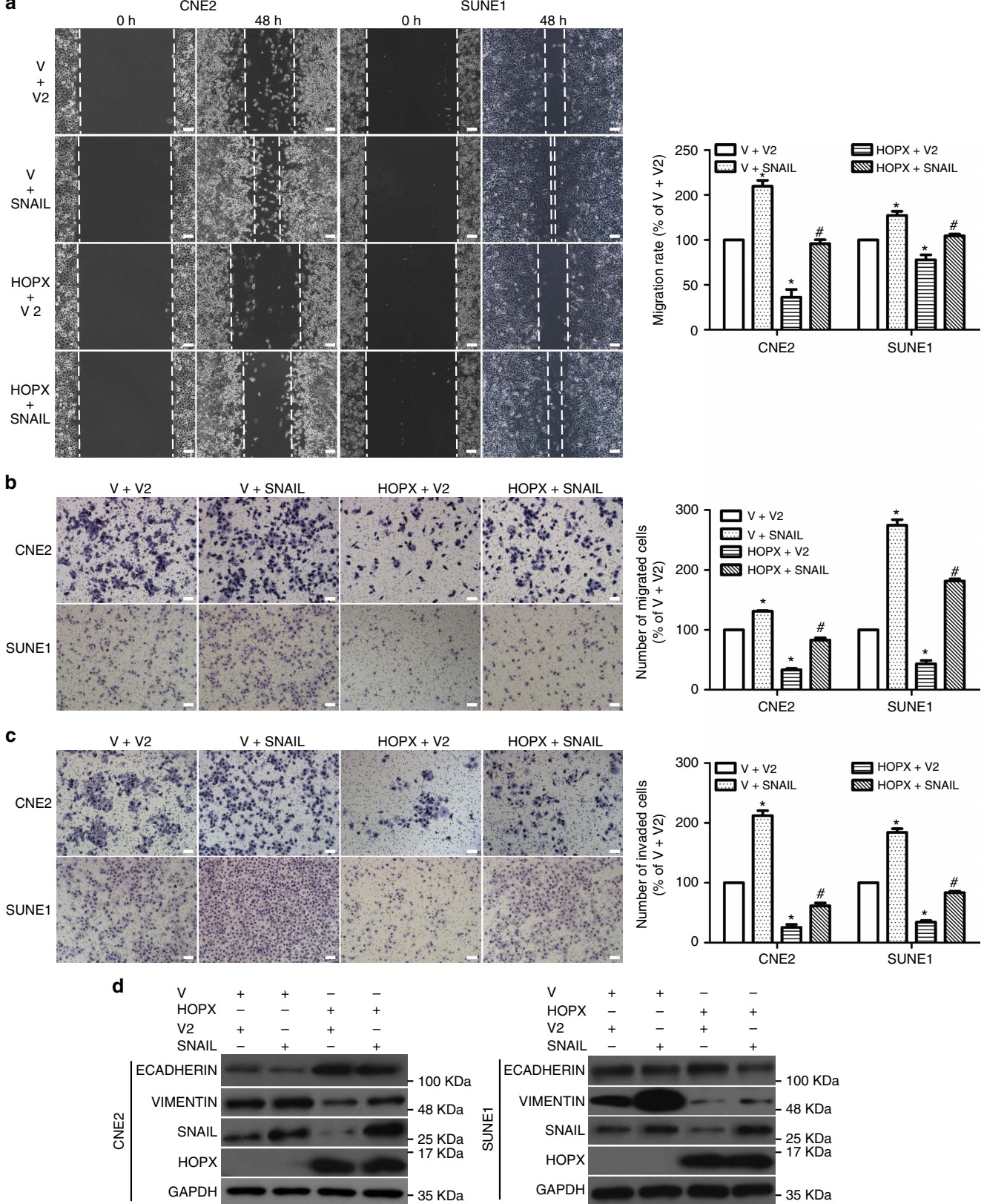

**Figure 7 | SNAIL is a functional target of HOPX in the regulation of EMT and metastasis of NPC cells.** Vector2 (V2) or *SNAIL* was transfected in CNE2 and SUNE1 cells that stably overexpressed the vector (V) or HOPX. (**a–c**) The migration and invasion abilities were measured using wound healing assays (×200) (**a**) and Transwell assays (×200) without (**b**) or with (**c**) Matrigel. Scale bar, 100 μm; mean ± s.d.; *P < 0.01 compared with V + V2; #P < 0.01 compared with the HOPX + V2; Student's *t*-tests. (**d**) ECADHERIN, VIMENTIN, SNAIL, HOPX and GAPDH expression levels were measured via western blotting. These data are representative of three independent experiments.

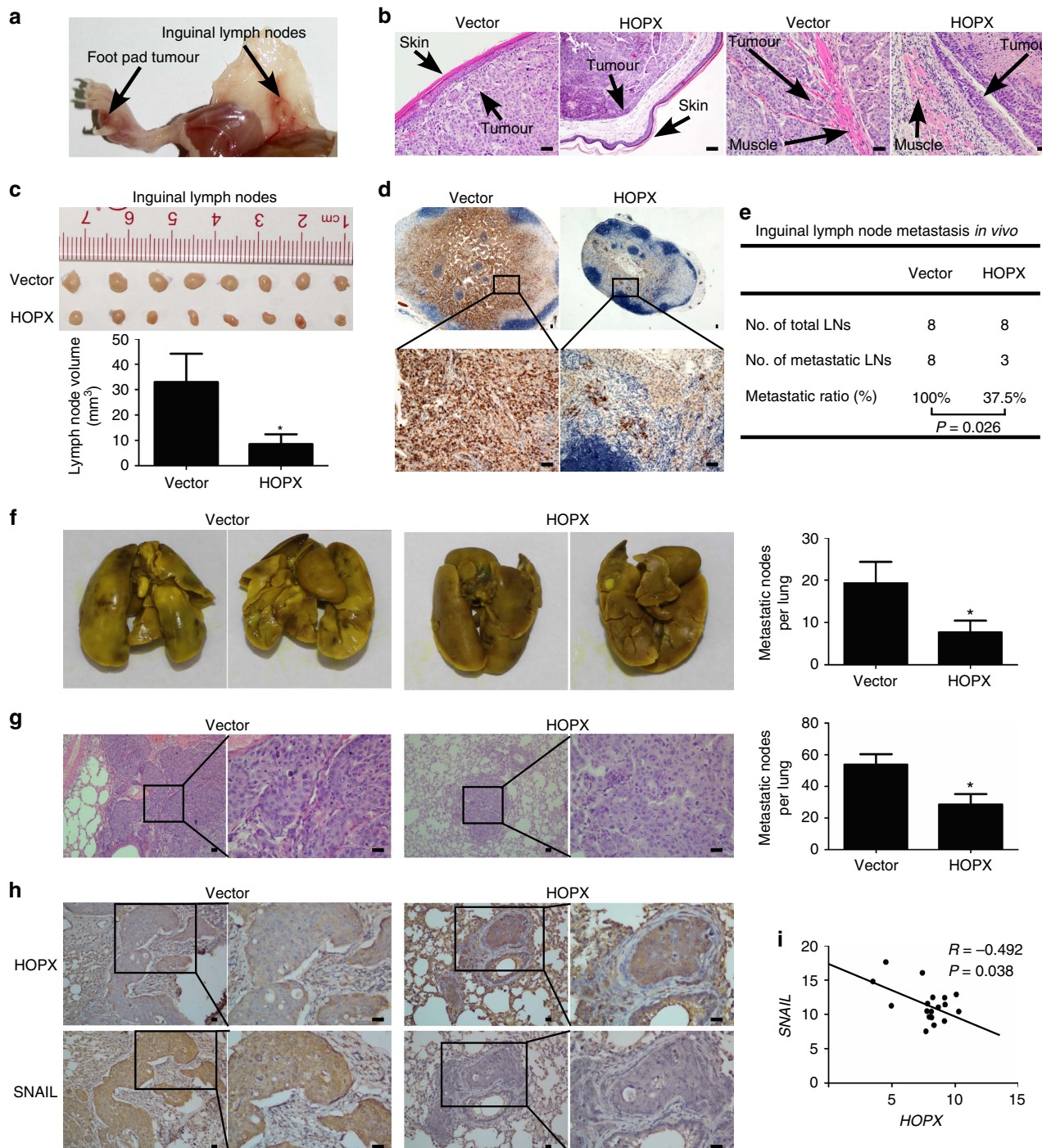

**Figure 8 | HOPX restoration inhibits NPC cell aggressiveness and *SNAIL* expression *in vivo*.** (**a–e**) SUNE1 cells stably overexpressed the vector or HOPX (*n* = 8 per group) were injected into the foot pads of mice to construct inguinal lymph node metastasis models. (**a**) Representative images of primary foot pad tumour and metastatic inguinal lymph node. (**b**) Representative images (×200) of microscopic primary tumours in foot pads stained with haematoxylin and eosin (H&E). Scale bar, 100 μm. (**c**) Representative images and quantification of the average volumes of the inguinal lymph nodes. Mean ± s.d.; *P < 0.01 compared with vector; Student's *t*-tests. (**d**) Immunohistochemical staining for pan-cytokeratin-positive tumour cells in inguinal lymph nodes (×40 and ×200). Scale bar, 100 μm. (**e**) Metastatic ratios of inguinal lymph nodes; Chi-square test was used for statistical analysis. (**f–h**) SUNE1 cells with vector or HOPX overexpression group (*n* = 5 per group) were injected into the tail veins of mice to construct a lung colonization model. (**f**) Representative images of macroscopic tumour colonization and growth in lung tissues and the number of tumour nodules in mice xenograft. Mean ± s.d.; *P < 0.01 compared with vector; Student's *t*-tests. (**g**) Representative images (×100 and ×400) and quantification of the average number of microscopic tumour nodes in lungs stained with H&E. Scale bar, 100 μm; mean ± s.d.; *P < 0.01 compared with vector; Student's *t*-tests. (**h**) Immunohistochemical staining for HOPX and SNAIL expression in the lungs of mice xenograft (×100 and ×400). Scale bar, 100 μm. (**i**) Relative *HOPX* and *SNAIL* expression values were determined via real-time RT–PCR in the NPC tissues (*n* = 24). Statistical analysis was performed using the Pearson's coefficient test. Each mouse sample was considered as an independent experiment; three technological replications were repeated in each sample.

tumour nodules (Fig. 8g). Collectively, these findings suggest that HOPX inhibits NPC cell lung colonization and metastasis *in vivo*.

To further examine whether HOPX suppressed SNAIL expression *in vivo*, IHC was used to assess HOPX and SNAIL protein levels in the lung sections. The results demonstrated that SNAIL expression was significantly decreased in the HOPX overexpression group compared with the vector group (Fig. 8h). Next, we investigated the correlation between HOPX and SNAIL in the clinical NPC tissues. *SNAIL* mRNA levels were detected in the same tissues used to determine the *HOPX* mRNA and methylation levels. Pearson correlation analysis indicated that *HOPX* expression inversely correlated with *SNAIL* expression (Fig. 8i), while the *HOPX* methylation level positively correlated with *SNAIL* expression in NPC tissue (Supplementary Fig. 12a). In summary, these results suggest that HOPX upregulation is associated with a reduction in *SNAIL* expression *in vivo* and in NPC clinical tissues.

**HOPX enhances the chemosensitivity of NPC cell to cisplatin**. Chemotherapy based on cisplatin (DDP) regimen is the standard therapy for advanced NPC patients[1]. However, chemoresistance is often associated with treatment failure. EMT has been considered to play an important role in chemoresistance in cancer cells[24,29]. To determine whether *HOPX* had any effect on chemosensitivity in NPC, the MTT assay was performed. Restoring HOPX expression significantly enhanced the sensitivity of NPC cells to DDP *in vitro* (Fig. 9a). To further investigate whether HOPX conferred chemosensitivity *in vivo*, a xenograft tumour model treated with normal saline or DDP was constructed. In the normal saline-treatment group, HOPX overexpression exhibited no significant effect on tumour growth compared with the vector (Fig. 9b,c,e). However, in the DDP-treatment group, the tumour volumes and weights were significantly suppressed following HOPX overexpression (Fig. 9b,d,e). Together, these findings imply that HOPX enhances the cisplatin sensitivity of NPC cells.

**HOPX methylation is related to survival in NPC patients**. To determine whether *HOPX* methylation was related to the clinical features of NPC patients, bisulfite pyrosequencing analysis was conducted on 443 NPC tissues from two hospitals. The clinical characteristics of the NPC patients in the training and validation cohorts are listed in Supplementary Table 2. To predict the outcomes of the NPC patients, the best cut-off values (*HOPX* low methylation: <13.5%; *HOPX* high methylation: ≥13.5%) for the low and high methylation levels of *HOPX* in the training cohort

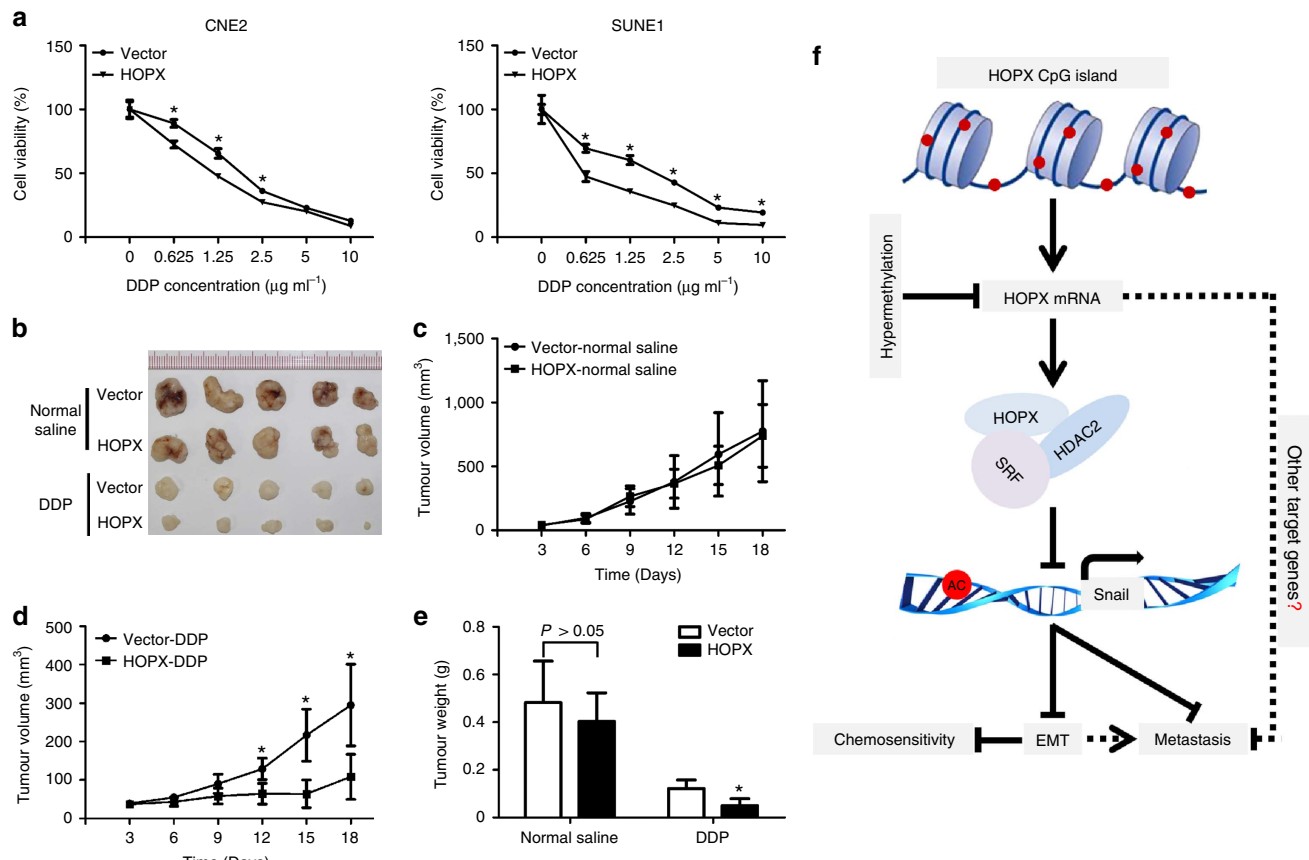

**Figure 9 | HOPX enhances the sensitivity of NPC cells to DDP.** (**a**) Dose–response curves of CNE2 and SUNE1 cells that stably overexpressed the vector or HOPX following DDP (0, 0.625, 1.25, 2.5, 5 and 10 μg ml⁻¹) treatment. Mean ± s.d.; *P<0.01 compared with vector; Student's *t*-tests. (**b–e**) CNE2 cells stably overexpressed the vector or HOPX were injected into the dorsal flank of the mice. The mice were randomly divided into four groups (n = 5 per group) and treated with normal saline or DDP (4 mg kg⁻¹). These data are representative of five independent experiments (each mouse sample was considered as an independent experiment). (**b**) Representative picture of xenograft tumours in nude mice. (**c,d**) The growth curves of the tumour volumes in normal saline (**c**) or DDP (**d**) treatment group. Mean ± s.d.; *P<0.01 compared with vector; Student's *t*-tests. (**e**) The tumour weights in normal saline- or DDP-treatment group. Mean ± s.d.; *P<0.01 compared with vector; Student's *t*-tests. (**f**) Schematic summary of the HOPX-HDAC2/SRF-*SNAIL* signalling pathway. HOPX suppresses metastasis and enhances chemosensitivity in NPC via the recruitment of histone deacetylase HDAC2 to epigenetically inhibit SRF-dependent *SNAIL* transcription. This suppression effect could partially release from the *HOPX* hypermethylation.

were selected on the basis of receiver operating characteristic curve analysis[36]. In the training cohort, 185 of 255 (72.5%) NPC patients showed representatively high *HOPX* methylation levels. The patients with high *HOPX* methylation exhibited worse distant metastasis-free survival (DMFS), overall survival (OS) and disease-free survival (DFS) compared with the patients with low *HOPX* methylation (Fig. 10a–c). To validate the survival prognostic accuracy of the *HOPX* methylation levels, the same cut-off values were applied to the validation cohort. The results indicated that 111 of 188 (59.0%) NPC patients in the validation cohort had representatively high *HOPX* methylation levels, and patients with high *HOPX* methylation also had shorter DMFS, OS and DFS (Fig. 10d–f).

We subsequently analysed the correlations between *HOPX* methylation levels and the clinical characteristics of the NPC patients. In the training cohort, *HOPX* methylation levels were associated with the TNM stage. No significant correlations between *HOPX* methylation levels and patient age, sex, World Health Organization type, viral capsid antigen immunoglobulin A or early antigen immunoglobulin A were identified. In the validation cohort, none of the clinical characteristics

differed significantly between the two groups was identified (Supplementary Table 2).

We also investigated whether the methylation levels of *HOPX* were associated with survival within TNM staging. First, we combined the NPC patients in the training and validation cohorts. Kaplan–Meier analysis indicated that high *HOPX* methylation was associated with worse DMFS, OS and DFS in the combined group (Supplementary Fig. 13a–c). We then classified all the patients into two groups (stage I–II and stage III–IV). Our findings indicated that stage III–IV patients with high or low *HOPX* methylation had similar DMFS, OS and DFS (Supplementary Fig. 14a–c).

To determine whether the methylation level of *HOPX* was an independent prognostic factor, multivariate Cox regression analyses were used. In the training cohort, the *HOPX* methylation level was identified as an independent prognostic factor for DMFS, OS and DFS (Supplementary Table 3). The results for the validation cohort and the combined cohort were the same (Supplementary Tables 3 and 4). Collectively, these findings imply that *HOPX* methylation levels correlated with the clinical outcomes of NPC patients.

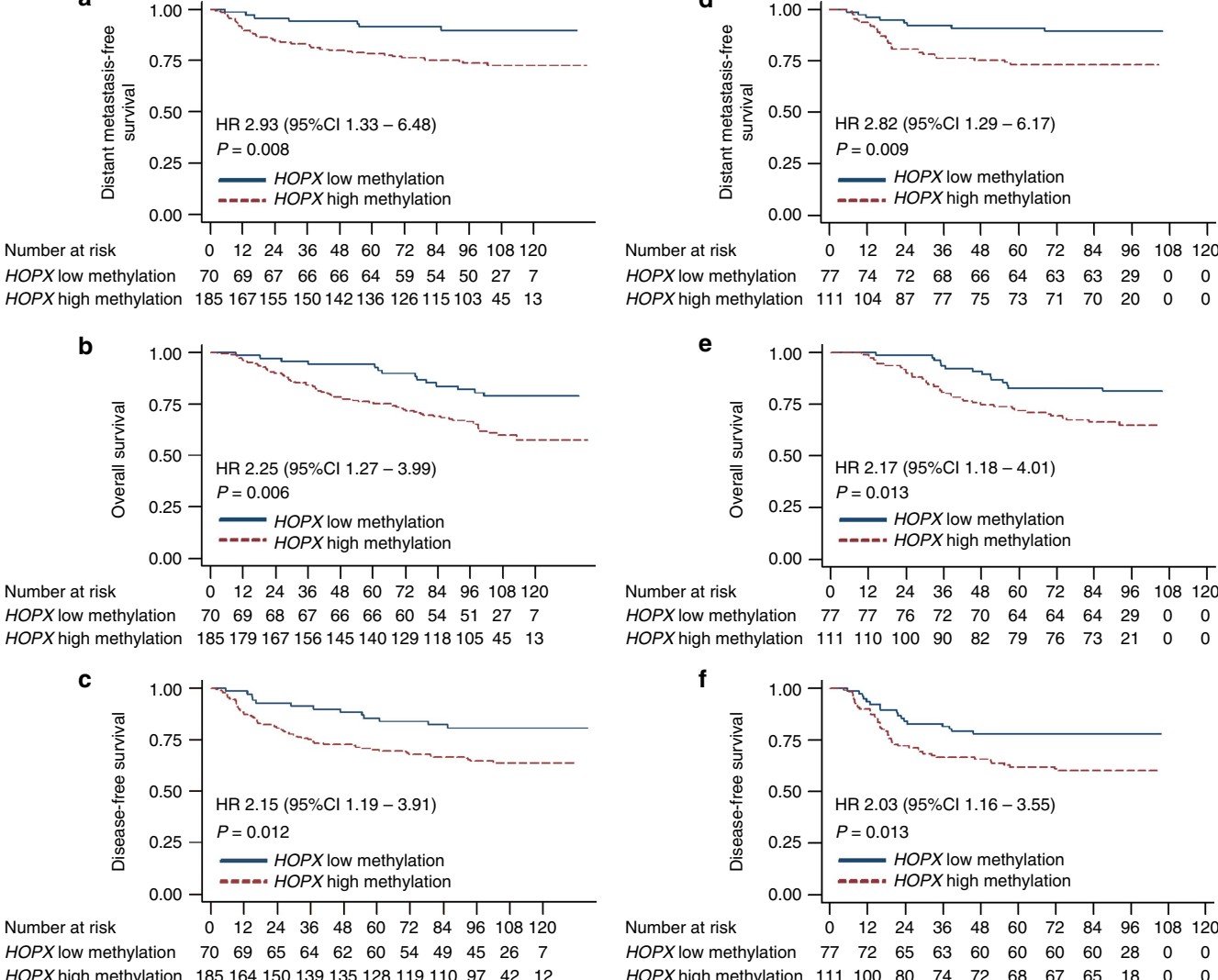

**Figure 10 | *HOPX* hypermethylation is associated with poor prognosis in NPC.** *HOPX* methylation levels were determined via bisulfite pyrosequencing analysis. Kaplan–Meier analysis was performed to determine the DMFS, OS and DFS according to the *HOPX* methylation level (low methylation versus high methylation) in NPC patients. (**a–c**) DMFS (**a**), OS (**b**) and DFS (**c**) in 255 NPC patients in the training set. (**d–f**) DMFS (**d**), OS (**e**) and DFS (**f**) in 188 NPC patients in the validation set. Adjusted univariate Cox proportional hazard models were used to calculate the HR values and *P* values.

## Discussion

This study identified novel roles for *HOPX* in NPC. Our findings demonstrated that, compared with normal nasopharyngeal epithelial tissues, the methylation levels of *HOPX* were significantly increased in NPC tissues. *HOPX* was downregulated in NPC as a result of its promoter hypermethylation. Restoration of HOPX significantly suppressed metastasis and enhanced chemosensitivity of NPC cells *in vitro* and *in vivo*. These HOPX repression effects might depend on the recruitment of the histone deacetylase HDAC2 and the enhancement of the histone H3K9 deacetylation of SRF-dependent *SNAIL* transcriptional. Moreover, patients with high *HOPX* methylation exhibited poor DMFS, OS and DFS.

Most cancers are caused by the accumulation of genomic or epigenetic alterations. Because gene mutation sites and frequency are rare and low in NPC[37,38]; epigenetic alterations are considered to play a vital role in pathogenesis[39,40]. Although several aberrantly methylated genes have been identified, our understanding of the metastatic progression of NPC remains inadequate. TFs are among the key elements in initiating and regulating gene expression. Specific groups of TFs that are abnormally expressed in many cancers have gained increased attention, such as steroid receptors (oestrogen receptors[41] and androgen receptors[42]), resident nuclear proteins (c-JUN[43]), and latent cytoplasmic factors (STATs[44] and NF-κB[45]). In this study, we focused on the abnormally methylated TFs in NPC. Compared with the normal nasopharyngeal epithelial tissues, *HOPX* is the most differentially hypermethylated TF in NPC tissues.

HOPX has a 60-amino-acid motif homologous to the homeodomain of Hox transcription factors and plays an important role in cell differentiation[46,47], proliferation[20] and migration[14]. Recently, *HOPX* has been demonstrated to be downregulated due to its promoter hypermethylation and acts as a tumour suppressor in several cancers, such as pancreatic carcinoma[48], uterine endometrial cancer[49], gastric cancer[20], colorectal cancer[21,50,51], esophageal squamous cell carcinoma[52] and lung cancer[19], implying that *HOPX* is likely to be a cancer-specific TF. However, *HOPX* has also been reported to be upregulated in thyroid cancer[53], sarcoma[54] and invasive pancreatic cancer[55], and increase tumour cell migration and invasion. In fact, many genes have been suggested to act as oncogenes or tumour suppressors in different cancer types[56,57]. There is little knowledge regarding the roles that *HOPX* plays in NPC. In this study, we demonstrated that *HOPX* was significantly downregulated in NPC patients (especially patients with metastasis) and cell lines as a result of its promoter hypermethylation. *HOPX* overexpression inhibited metastasis and enhanced sensitivity to cisplatin of NPC cells, and did not exert a significant influence on cell proliferation ability. Therefore, we conclude that *HOPX* acts as a tumour suppressor mainly with regard to NPC metastasis and chemosensitivity.

Despite substantial evidence indicating that *HOPX* is associated with multiple cancers, the mechanism of *HOPX* in cancer remains elusive. In lung adenocarcinomas, *GATA6* and *HOPX* play critical roles in a lineage-selective pathway to suppress metastasis. The simultaneous knockdown of these factors causes cells to exhibit characteristics of basal epithelial cells; however, this change is not sufficient to induce EMT and has little effect on the WNT signalling pathway[14]. During the development of the mouse cardiovascular system, HOPX physically interacts with SMAD4 to repress WNT signalling. The authors assume that this signalling pathway may underlie the tumour-suppressor function of *HOPX*[34]. Our findings also indicated that HOPX physically interacted with SMAD4; however, there was little effect on the canonical WNT signalling pathway (Supplementary Fig. 15a,b).

Interestingly, after silencing HOPX expression, the morphology of some NPEC cells partially transitioned to a mesenchymal type. A tumour metastasis PCR array showed that the expressions of many genes (*ECADHERIN*, *NME4*, *SYK*, *CD44* and *IL1B*) were changed in NPC cells with HOPX overexpression. HOPX significantly reduced the expression of SYK, which was associated with poor clinical outcomes of NPC patients[58]. However, CD44 that could promote NPC progression was also upregulated by HOPX[59], suggesting that HOPX exerts a broad effect on NPC cells. Notably, *ECADHERIN* which might be responsible for the morphology changes of NPC cells induced by HOPX was upregulated. Furthermore, the restoration of HOPX promoted epithelial markers expression and inhibited mesenchymal markers expression, indicating that *HOPX* partly represses EMT in NPC cells *in vitro*.

EMT, which can be triggered by EMT-TFs, has been commonly thought to be a crucial process for epithelial tumour cells to dissociate and disseminate to distant sites, as well as to confer chemoresistance. Although there are two studies with provocative results challenging the importance of EMT in metastasis[24,29], the involvement of EMT in cancer progression cannot yet be eliminated[60–62]. *SNAIL* is among the most important EMT-TFs and has been demonstrated to exhibit a broad effect on embryonic development and tumour progression[63]. Not surprisingly, *SNAIL* also has a vital effect on NPC progression. High nuclear SNAIL expression has been reported to be associated with poor survival in NPC[64]. Numerous regulators exert their functions on NPC progression through regulating *SNAIL*, such as BMI-1 (ref. 65), EZH2 (ref. 66), *LMP1* (ref. 67), *EBV-mir-BART7-3p* (ref. 68) and *ME1* (ref. 69). BMI-1 has been reported to induce EMT via stabilizing SNAIL through the PI3K–AKT–GSK3β signalling pathway in NPC cells, ultimately enhancing the lung metastatic colonization ability of tumour cells[65]. EZH2 can form a complex with HDAC1, HDAC2 and SNAIL to enhance NPC cells metastasis *in vitro* and lung metastatic colonization *in vivo*[66]. Therefore, a comprehensive understanding of the regulatory mechanism of *SNAIL* expression will provide vital information regarding EMT blockage and tumour suppression in NPC. In this study, after screening the EMT-TFs affected by HOPX, *SNAIL* was demonstrated to be the most significantly suppressed EMT-TF. HOPX could inhibit EMT via binding to the promoter region of *SNAIL* and inhibiting its transcription.

HOPX acts as a corepressor via the recruitment of HDACs in the myocardial development of mice and zebrafish[12,13]. However, the mechanism by which HOPX represses target genes in human cancer is unknown. Here, based on mass spectrometry, we found that HOPX could recruit and interact with HDAC2, which then deacetylated histone H3K9 to epigenetically silence *SNAIL* transcription in NPC cells. Following the search for the *SNAIL* promoter region that bound with HOPX and the HOPX-interacting proteins identified by mass spectrometry, we demonstrated that HOPX suppressed the interaction of SRF with the *SNAIL* promoter and epigenetically inhibited SRF-dependent *SNAIL* transcription. The restoration of SNAIL in NPC cells with HOPX overexpression could significantly reverse the inhibitory effects of HOPX on metastasis and EMT, indicating that *SNAIL* is a functional target of HOPX in NPC.

Appropriate treatment choices are based on accurate prognostic assessments. HOPX is also associated with the prognosis of colorectal adenocarcinoma[50], lung cancer[19] and stomach cancer[20], which suggests that HOPX might represent a promising biomarker for the prediction of malignant diseases outcomes. To date, metastasis is the major mode of treatment failure in NPC patients. There is no effective forecasting model that can accurately select patients with a high risk of metastasis.

In this study, we demonstrated that aberrant methylation of *HOPX* promoter exhibited an essential role in NPC metastasis; thus, we analysed and validated the methylation status of *HOPX* with regard to NPC clinical outcomes. We demonstrated that patients with high methylation levels of *HOPX* exhibited poor DMFS, OS and DFS. *HOPX* hypermethylation was an independent prognostic biomarker for NPC patients. Therefore, the examination of *HOPX* methylation levels might represent a promising approach for identifying subgroups of NPC patients with an underlying poor prognosis, which might facilitate the selection of more appropriate individual therapies for these patients.

In summary, this research highlighted the importance of a newly identified signalling pathway triggered by *HOPX* (Fig. 9f). As a key molecular component of this axis, HOPX could epigenetically inhibit SRF-dependent *SNAIL* transcription via the recruitment of HDAC2 and the enhancement of histone H3K9 deacetylation, which subsequently suppressed NPC progression. This suppression effect could partially release from the *HOPX* hypermethylation. *HOPX* methylation levels may serve not only as a prognostic factor to predict the clinical prognosis of NPC patients but also as a therapeutic target.

## Methods

**Microarray analysis.** Infinium Human Methylation 450 K BeadChip microarray data was extracted and analysed to examine the methylation levels of TFs. GenomeStudio methylation module V.1.9.0 (Illumina, USA) was used to extract the image intensities. The CpG probe signal intensities were normalized using a quantile algorithm in the 'methylumi' and 'lumi' packages from R Bioconductor for background correction and subtraction. The normalized signal intensities were used to calculate $\beta$-values, which represent the DNA methylation levels for per-probe loci, as indicated in our previous report (GSE52068; ref. 30). Briefly, $\beta$-value = methylated/(methylated + unmethylated + 100) (range: 0–1). Significantly different methylation loci were screened using the 'IMA' Bioconductor package (PMID: 22253290). Single-nucleotide polymorphism probes, probes with a detection $P$ value $\geq 0.01$ in 12 of 48 samples and probes located in the X and Y chromosomes were removed. Ultimately, 473,788 of the original 485,577 CpG sites were retained. The significantly different DNA methylation loci between the NPC and normal tissues were further identified using a nonparametric Wilcoxon rank-sum test with a false discovery rate adjustment. A significantly different locus was defined by a threshold of adjusted $P < 0.05$ and $\Delta\beta$ change $\geq |0.2|$. In this study, only the CpG sites within the promoter (TSS and 5′-UTR) regions of TFs (http://www.bioguo.org/AnimalTFDB/) were retained for analysis.

**Clinical specimens.** This study was approved by the Institutional Ethical Review Board of the participating hospitals. From 2004 to 2007, we collected 443 formalin-fixed paraffin-embedded (FFPE) NPC tissues from the primary site, with detailed long-term follow-up clinical data from the Sun Yat-sen University Cancer Center ($n = 255$, Guangzhou, China) and Zhejiang Cancer Hospital ($n = 188$, Hangzhou, China). All the samples contained >70% tumour cells based on H&E staining, examined and validated by two authoritative pathologists (Dr Junping Yun, Sun Yat-sen University Cancer Center, China; and Dr Jian Zeng, Zhejiang Cancer Hospital, China). The seventh edition of the American Joint Committee on Cancer (AJCC) Cancer Staging Manual was applied to redefine the TNM staging. All the patients provided written informed consent for the use of their biopsies. None of the patients had received anti-tumour treatment before biopsy collection. Following tissue collection, all the patients were treated with definitive-intent radiotherapy; the patients with stage III–IV also received platinum-based concurrent chemotherapy during radiotherapy. All the patients received regular post-radiotherapy clinical assessment, and the median follow-up time was 94 months (range: 2–139 months). This study followed the REMARK guidelines (REporting recommendations for tumour MARKer prognostic studies).

**Cell culture.** Keratinocyte serum-free medium (Invitrogen, USA) supplemented with bovine pituitary extract (BD Biosciences, USA) was used to grow the three NPEC cell lines (N2-Tert, NP69 and N2-Bmi1). Human NPC cell lines (SUNE1, CNE1, CNE2, HONE1 and HNE1) were maintained in RPMI 1640 (Invitrogen) supplemented with 10% fetal bovine serum (FBS, Gibco, USA). All NPEC and NPC cell lines were generously provided by Professor Musheng Zeng (Sun Yat-sen University Cancer Center, China). 293FT cells obtained from ATCC were cultured in Dulbecco's modified Eagle's medium (Invitrogen) supplemented with 10% FBS.

**DAC and TSA treatment.** For the methyltransferase inhibitor DAC (Sigma-Aldrich, USA) treatment, $1.5 \times 10^5$ cells were seeded on 60 mm culture dishes. After growing for 24 h, the cells were treated with or without DAC (10 µM) by replacing the drug every 24 h for 72 h. For the HDAC inhibitor TSA (Sigma-Aldrich) treatment, $3 \times 10^5$ cells were seeded on 60 mm culture dishes. Following 24 h growth, the cells were treated with TSA (0, 50 or 100 nM) for 48 h. After that, the cells were collected for DNA or RNA extraction.

**DNA extraction and bisulfite pyrosequencing analysis.** Genomic DNA from FFPE, fresh-frozen tissues and cells was isolated using a QIAamp DNA FFPE Tissue Kit (Qiagen, Germany), AllPrep RNA/DNA Mini Kit (Qiagen) or EZ1 DNA Tissue Kit (Qiagen), respectively, according to the manufacturer's instructions. An EpiTect Bisulfite Kit (Qiagen) was applied to conduct the bisulfite modification of DNA (1–2 µg). PyroMark Assay Design Software 2.0 (Qiagen) was used to design the bisulfite pyrosequencing primers. The PyroMark Q96 ID System and software (Qiagen) were utilized for the sequencing reaction and methylation level quantification. The primer sequences for PCR and sequencing are shown in Supplementary Table 5.

**Real-time RT–PCR and tumour metastasis PCR array.** TRIzol reagent (Invitrogen) was used to isolate total RNA from NPC cells and clinical tissues. Real-time RT–PCR was performed to measure the target gene mRNA levels. A tumour metastasis PCR array (Qiagen) was performed according to the manufacturer's instructions. Briefly, first-strand complementary DNA was synthesized using random primers (Promega, USA) and M-MLV reverse transcriptase (Promega). A CFX96 Touch sequence detection system (Bio-Rad, USA) was used to conduct the SYBR Green-based (Invitrogen) real-time PCR analysis. *GAPDH* was considered an endogenous control for all the genes. The relative gene expression was calculated using the comparative threshold cycle ($2^{-\Delta\Delta CT}$) equation. All the experiments were performed in triplicate, and the primer sequences are shown in Supplementary Table 5.

**IHC assay.** IHC was performed on FFPE sections of clinical NPC tissues or xenograft mice tissues. Briefly, the tissues were deparaffinized and rehydrated; the endogenous peroxidase activity was blocked; and the samples were subjected to citrate-mediated high-temperature antigen retrieval. The nonspecific binding was subsequently blocked, and the samples were incubated with the primary antibodies at 4 °C overnight. All the sections were scored and validated by the two experienced pathologists. The expression level was calculated using the following equation: staining index = staining intensity × percentage of positive cells. The staining intensity was defined as following: 0, no staining; 1, weak, light yellow; 2, moderate, yellow brown; and 3, strong, brown. The proportion of positive cells was defined as follows: 1, <10%; 2, 10–35%; 3, 35–70%; 4, >70%. The antibodies used for IHC assays are shown in Supplementary Table 6.

**Western blotting assay.** Total protein was isolated using RIPA buffer (Beyotime Biotechnology, China) that contained a protease inhibitor cocktail (FDbio Science, China). Protein extracts were separated via 8–12% sodium dodecyl sulfate-polyacrylamide gel electrophoresis and transferred to polyvinylidene fluoride membranes (Millipore, USA). The membranes were subsequently blocked in 5% defatted milk and incubated with primary antibodies overnight at 4 °C. The species-matched secondary antibodies were then hybridized with the membranes at room temperature. Finally, the antigen–antibody reaction was visualized using enhanced chemiluminescence (Thermo, USA). The antibodies used for western blotting assays are shown in Supplementary Table 6. Uncropped western blots can be found in Supplementary Fig. 16.

**RNA interference and plasmid transfection.** Effective siRNA oligonucleotides that targeted *HOPX* and *HDAC2* were purchased from GenePharma (China). The pSin-EF2-puro-Vector, pSin-EF2-puro-*HOPX*, pLEGFP-N1, pLEGFP-N1-*SNAIL* and pLEGFP-N1-*SRF* plasmids were obtained from Vigene Bioscience (China) and Long Bioscience (China). All the plasmids were verified by DNA sequencing. The pSin-EF2-puro-Vector and pLEGFP-N1 plasmids were used as the controls. The siRNA sequences for *HOPX* are shown in Supplementary Table 5.

To generate stably transfected cell lines, Lentivirus packing expression plasmids were co-transfected into 293FT cells. The supernatants that contained viruses were subsequently infected with NPC cells for 48 h. Following infection, the stable clones were selected with 0.5 µg ml$^{-1}$ puromycin (Sigma-Aldrich) or/and 1 mg ml$^{-1}$ G418 (Sigma-Aldrich). The infection efficiency was validated using real-time RT–PCR or western blotting assays.

For transient transfection, Lipofectamine 2000 reagent (Invitrogen) or Lipofectamine RNAiMAX transfection reagent (Invitrogen) was used according to the manufacturer's instructions. The cells were collected after transfection with siRNA oligonucleotides (100 nM) or plasmids (2 µg) for 48 h.

**Wound healing assay.** The cells grown to near confluence in six-well plates were subjected to serum-free medium for 24 h of starvation. The monolayers were scratched using a sterile 200 µl tip, followed by an additional 48 h of starvation.

An inverted microscope (Olympus IX73, Japan) was used to capture photos of the cells migrating at the corresponding wound sites at 0 and 48 h.

**Transwell migration and invasion assays.** The cell migration and invasion assays were performed using Transwell chambers (8 µm pores, Corning, USA) pre-coated without (migration assay) or with (invasion assay) Matrigel (BD Biosciences). First, $5 \times 10^4$ or $1 \times 10^5$ cells suspended in 200 µl of serum-free medium were plated in the upper chambers, whereas 500 µl of medium supplemented with 10% FBS was placed in the lower chambers. Following 12 h (migration assay) or 24 h (invasion assay) of incubation, the cells on the upper surface of the membrane filter were fixed with methyl alcohol, stained with haematoxylin and counted under an inverted microscope.

**MTT and colony-formation assays.** For the cell viability assay, $1 \times 10^3$ cells in 200 µl medium were seeded per well in 96-well plates. Twenty microlitres MTT ($5 \, mg \, ml^{-1}$, BD Biosciences) were added per well following incubation for the indicated time periods (1, 2, 3, 4 and 5 days). Then, the supernatants were discarded and 150 µl dimethyl sulfoxide was added to each well. A spectrophotometric plate reader (BioTek ELX800, USA) was used to measure the absorbance at 490 nm. For DDP treatment, $1 \times 10^3$ cells in 200 µl medium were seeded per well in 96-well plates for 24 h incubation. Then, the cells were treated with DDP (0, 0.625, 1.25, 2.5, 5 and $10 \, \mu g \, ml^{-1}$) for 72 h. After that, the cell viability was measured.

For the colony-formation assay, 400 cells in 2 ml of medium were plated per well in six-well plates. Following 7 (CNE2 cells) or 12 (SUNE1 cells) days of incubation, the colonies were quantified after fixation with methyl alcohol and staining with haematoxylin. These experiments were performed three times.

**Immunofluorescence assay.** For immunofluorescent staining, the cells were fixed in methyl alcohol, permeabilized in phosphate-buffered saline (PBS) with 0.5% Triton X-100, and incubated with the primary antibodies. The cells were subsequently stained with species-matched secondary antibodies. Nuclei were counterstained with DAPI (4′, 6-diamidino-2-phenylindole; Sigma-Aldrich), and the slides were viewed using a confocal laser-scanning microscope (Olympus FV1000, Japan). The antibodies used for immunofluorescent assays are shown in Supplementary Table 6.

**Luciferase reporter assay.** For the luciferase reporter assays, psiCHECK luciferase reporter plasmids (Promega) that contained the wild-type and mutant of the SNAIL promoter were constructed. Then, $5 \times 10^4$ cells per well seeded in 24-well plates were co-transfected with pGL3-basic using Lipofectamine 2000 reagent (Invitrogen) for 36 h. Eight nanograms of pRL-CMV (Renilla luciferase) per well were co-transfected to normalize the transfection efficiency. Passive Lysis Buffer (Promega) was used to collect the cells, and the Dual Luciferase Reporter Assay System (Promega) was used to detect the luciferase activity.

**CHIP assay.** CNE2, SUNE1, NP69 and N2-Tert cells were applied to the ChIP assays. An EZ-Magna ChIP kit (Millipore) was used to perform the ChIP assay according to the manufacturer's protocol. Briefly, 1% formaldehyde solution was applied to the cells to induce crosslinking, followed by quenching crosslinking with 140 mM glycine. The nucleoprotein complexes were subsequently lysed and sheered to 200–500 bp yield DNA fragments and immunoprecipitated with the antibody or IgG (negative control) overnight at 4 °C. After the DNA was uncrosslinked, PCR and real-time PCR analyses were performed to the regions of interest. For the ChIP real-time PCR assay, the amount of immunoprecipitated DNA was normalized to the input. The primers used to amplify the SNAIL promoter regions are shown in Supplementary Table 5. The antibodies used for the ChIP assays are shown in Supplementary Table 6.

**Mass spectrometry and Co-IP assay.** For the IP assay, the cells were lysed with IP lysis buffer. Primary anti-HOPX or anti-IgG (negative control) antibodies were incubated with the lysates overnight at 4 °C. Protein A/G Sepharose beads (Santa Cruz, USA) were added to the immune-complexes for recovery. After that, the immune-complexes were washed and collected. Mass spectrometry was conducted by Huijun Biotechnology (China). For Co-IP assay, western blotting was applied to determine the protein levels of interest. The antibodies used for the IP assays are shown in Supplementary Table 6.

**In vivo xenograft tumour models.** All animal research procedures were performed according to the detailed rules of the Animal Care and Use Ethics Committee of Sun Yat-Sen University Cancer Center. We did our best to minimize animal suffering. The BALB/c nude mice (4–5 weeks old, female) bred and maintained at the Animal Experiment Center of Sun Yat-Sen University were obtained from the Medical Experimental Animal Center of Guangdong Province (China).

For the inguinal lymph node metastasis model, $2 \times 10^5$ per 30 µl of PBS of SUNE1 cells that stably overexpressed the vector or HOPX were injected into the foot pads of the mice (n = 8 per group). Following 6 weeks of growth, the mice were killed. Their foot pad tumours and inguinal lymph nodes were detached.

To establish a xenograft tumour model of lung colonization, $1 \times 10^6$ per 200 µl of PBS of SUNE1 cells that stably overexpressed the vector or HOPX were injected into the tail veins of the mice (n = 5 per group). After 6 weeks of growth, the mice were killed, and their tumour xenografts, the lung tissues, were harvested. All the foot pad tumours, inguinal lymph nodes and lung tissues were paraffin-embedded and cut into 5 µm tissue sections for subsequent analysis. The harvested foot pad tumours and lung tissue sections were stained with H&E for histological validation. The inguinal lymph nodes were incubated with an anti-pan-cytokeratin antibody (Thermo) for IHC analysis and evaluated by the pathologists.

The tumour growth model was constructed by injecting CNE2 cells that stably overexpressed the vector or HOPX into the dorsal flank of the mice. After 3 days of growth, the tumour nodes became palpable (approximately 100 mm³). Then, the mice were randomly divided into four groups (n = 5 per group) and injected intraperitoneally with normal saline or DDP every 3 days: Vector + Normal saline; HOPX + Normal saline; Vector + DDP ($4 \, mg \, kg^{-1}$); HOPX + DDP ($4 \, mg \, kg^{-1}$). Tumour size was measured every 3 days for approximately 2 weeks. Then, the mice were killed, and the tumours were dissected and weighted.

**Statistical analyses.** The SPSS 16.0 software (SPSS Inc., USA) was used for all statistical analyses, and a P value < 0.05 was considered significant. All data presented as the mean ± s.d. were extracted from no less than three independent experiments. The $\chi^2$ or Fisher exact tests were used for categorical variables. The Z-score method was used to quantify the methylation data examined by pyrosequencing. The equation used for the Z-score method was defined as follows: (methylation level of each gene in each sample − mean methylation level of each gene among all samples)/s.d. of methylation level for each gene. Stata 10 software was used for survival-associated statistical analyses. The Kaplan–Meier method and univariate analysis were used to estimate the survival curves, and multivariate Cox regression analysis with the backward stepwise method was used to determine the independent prognostic factors. To compare the differences between two independent groups, Student's t-tests were used. The correlation between HOPX and SNAIL was analysed using Pearson's coefficient test.

**Data availability.** The Genome-wide methylation microarray data referenced in this study have been previously deposited in the NCBI GEO database under accession code GSE52068. All other data are included in this published article or are available from the corresponding authors on reasonable request.

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

## Acknowledgements

We thank Hanqi Yin (Shanghai Biotechnology Corporation, China) and Qi Zhao (Sun Yat-sen University Cancer Center, China) for statistical consultation. This work was supported by grants from the Young Teachers Cultivation Project of Sun Yat-sen University from the Fundamental Research Funds for the Central Universities (No. 16ykpy21, N.L.); the National Natural Science Foundation of China (No. 81572658, J.M.); the Science and Technology Project of Guangzhou City, China (No. 14570006, J.M.); the Health and Medical Collaborative Innovation Project of Guangzhou City, China (No. 201400000001, J.M.); the Planned Science and Technology project of Guangdong Province (No. 2013B020400004, J.M.); and the National Science and Technology Pillar Program during the Twelfth Five-year Plan Period (No. 2014BAI09B10, J.M.).

## Author contributions

J.M., N.L., X.R., X.Y., B.C. and X.C. designed the research. X.R., X.Y., T.Z., Q.H., Y.L., X.T. and X.W. conducted the experiments. X.R., X.Y., T.Z., Q.H., Y.L., X.T. and X.W. acquired the data. X.R., X.Y., B.C., B.L., Q.Z., T.K., M.Z. and N.L. analysed the data. J.M., N.L., X.C., B.L., B.C. and M.Z. provided the reagents. X.R., X.Y., B.C., X.C., Q.Z., T.K., M.Z., N.L. and J.M. wrote the manuscript.

**Additional information**

**Competing financial interests:** The authors declare no competing financial interests.

