## [Peer Review File · Nature Communications]

Reviewer #1 (Remarks to the Author)

The manuscript by Ren et al describes the molecular mechanisms dictating the transformation and metastasis of Nasopharyngeal carcinoma (NPC). Global analysis of DNA methylation defects led to identification of HOPX transcription factor (TF) among the most highly methylated TFs. Authors then follow the HOPX downstream mechanisms, which may influence NPC outcomes. SNAIL was found as a primary target gene, where HOPX cooperates with HDAC2 and SRF to repress it, leading to inhibition of expression of EMT markers and reduced cell migration and invasion phenotypes. In vivo lung tumour colonisation experiments demonstrate lower number of metastatic nodes appearing from cells overexpressing HOPX.

HOPX tumour suppressor function was demonstrated before in other cancers. The role in epithelial to mesenchymal transition (EMT) was suggested as well. Thus the main novelty of the paper comes from documenting HOPX involvement in NPCs and the EMT via direct regulation of SNAIL. The manuscript convincingly demonstrated that the cellular phenotypes produced by reduced expression of HOPX are likely to be caused by activation of SNAIL, however it failed to convince that the HOPX directly regulates SNAIL. The details are discussed below:

1. The main weakness of the argument is the lack of unbiased techniques to support the main mechanism. For example, the molecular complex repressing transcription of SNAIL (HOPX, HDAC2, SRF) is identified by co-IPs of few selected candidates. This finding could be substantially stronger if unbiased IP-MS approach was used and the HDAC2 and SRF were found among the strongest interactors. Moreover, it is unclear if HOPX regulates other genes in addition to SNAIL, which could impact the phenotype of NPC. This could be addressed if the effects of HOPX overexpression measured by microarray or HOPX ChIP-seq was performed in cells, which naturally express HOPX. By selected examination of few factors it is difficult to build more complete picture of the roles of HOPX in NPCs.
2. It is surprising that around 20% elevated DNA methylation between the cell lines correlates with nearly 150 fold decrease in expression. Since only 20% alleles in the population of cells is methylated, even if methylated allele would silence the gene completely, it is difficult to explain such a dramatic reduction in expression. This implies that other silencing or defective activation mechanism is responsible for the downregulation of HOPX expression.
3. The direct interactions between HOPX and HDAC2, SRF seems very weak if any (considering the strength of IgG signals). Endogenous co-IPs from HOPX expressing cell lines (N69 and N2-Tert) should be produced to convince that the interactions are real.
4. Fig 1d, Fig 2f,g .Methylation of HOPX in cell lines should be shown as actual values instead of relative to NP69. The actual values are important for interpretation of expression data in fig 2d. Since experiments were done using pyrosequencing, the values should be readily available.
5. Fig 5f ChIP experiments should be quantified with quantitative PCR techniques.

Reviewer #2 (Remarks to the Author)

In this manuscript, Ren et al observe that promoter hypermethylation lead to the inactivation of HOPX in NPC. Restoration of HOPX inhibits NPC cell migration, invasion, and metastasis through recruiting HDAC2 suppressing SNAIL-mediated EMT. Finally, they analyze HOPX promoter hypermethylation is correlated with clinical features and poor prognosis of NPC patients. Their results demonstrate the significance of HOPX in NPC. Unfortunately, the biggest problem for their paper is that their core data markedly lack of original innovation for mechanism study in key points.

Key Comments:

1. HOPX as a tumor suppressor suppressing cell proliferation, migration, and invasion has been reported in some tumors including pancreatic cancer, uterine endometrial cancer, and gastric cancer, colorectal cancer, lung cancer et al., which is attributed to its promoter hypermethylation. Thus, promoter hypermethylation of HOPX in NPC is not a novel finding.

2. In this study, the observation about "HOPX epigenetically suppresses SNAIL transcription via HDAC2 recruitment" should be the most core result for this paper. The authors found HOPX epigenetically suppresses SNAIL transcription via recruiting HDAC2, but the interactions for HOPX with HDAC2 (Cardiac hypertrophy and histone deacetylase-dependent transcriptional repression mediated by the atypical homeodomain protein Hop. *J Clin Invest.* 2003; 112(6):863-71; Hopx and Hdac2 interact to modulate Gata4 acetylation and embryonic cardiac myocyte proliferation. *Dev Cell.* 2010; 19(3):450-9.) as well as HDAC2 with SNAIL (Repression of 15-hydroxyprostaglandin dehydrogenase involves histone deacetylase 2 and snail in colorectal cancer. *Cancer Res.* 2008; 68(22):9331-7; E-cadherin regulates metastasis of pancreatic cancer in vivo and is suppressed by a SNAIL/HDAC1/HDAC2 repressor complex. *Gastroenterology.* 2009; 137(1):361-71; Snail mediates E-cadherin repression by the recruitment of the Sin3A/histone deacetylase 1 (HDAC1)/HDAC2 complex. *Mol Cell Biol.* 2004; 24(1):306-19; Hepatitis C virus core protein interacts with Snail and histone deacetylases to promote the metastasis of hepatocellular carcinoma. *Oncogene.* 2015 Nov 9. [Epub ahead of print]; EZH2 supports nasopharyngeal carcinoma cell aggressiveness by forming a co-repressor complex with HDAC1/HDAC2 and Snail to inhibit E-cadherin. *Oncogene.* 2012; 31(5):583-94; HDAC2 provides a critical support to malignant progression of hepatocellular carcinoma through feedback control of mTORC1 and AKT. *Cancer Res.* 2014 ; 74(6):1728-38; A novel domain in histone deacetylase 1 and 2 mediates repression of cartilage-specific genes in human chondrocytes. *FASEB J.* 2009; 23(10):3539-52) had been reported. Furthermore, the interaction of HDCA2 with SRF was also reported (Homeobox gene HOPX is epigenetically silenced in human uterine endometrial cancer and suppresses estrogen-stimulated proliferation of cancer cells by inhibiting serum response factor. *Int J Cancer.* 2009; 124(11):2577-88.). Finally, the research about "SRF modulated Snail via binding to its promoter" had also been documented (Serum response factor accelerates the high glucose-induced Epithelial-to-Mesenchymal Transition (EMT) via snail signaling in human peritoneal mesothelial cells. *PLoS One.* 2014; 9(10):e108593.)

Other comments:

1. Suppressing HOPX in the human immortalized normal nasopharyngeal epithelial cell (NPEC) lines induces the to a spindle-shaped or elongated, mesenchymal form, why NPC cells with reduced or absent HOPX expression do not indicate this change?
2. NPC is an EBV-related tumor. LMP1 a critical EBV oncogenic gene in NPC. Whether LMP1 modulated LMP1 expression? If yes, How about the mechanism?
3. In fig.3 and fig.4, HOPX expression levels did not be tested in knockdown HOPX cells. Why you did not examine the expression change?
4. In fig.5f for CNE2 ChIP, No obvious difference was observed for snail DNA level in IgG and HOPX lanes. Thus, EMSA should be used to confirm the combination of HOPX with Snail via recruiting HDAC2;
5. Although the author confirm the interaction of HOPX with HDAC2 in NPC, they did not confirm whether HOPX modulated the expression of HDCA2? If HOPX modulated the expression of HDCA2, please clarify the molecular basis.
6. The author pointed out that HOPX interacted with SRF by endogenous CoIP. However, GST-PULL Down was needed to further confirm the interactions between HOPX with SRF.
7. Endogenous immuno-co-location should be performed among HOPX, HDCA2, and SRF.
8. In the HOPX-overexpressed NPC cells, the author should modulated the HDCA2 expression and observed the changes of Snail and its -mediated EMT, including EMT function and molecular mechanism (protein level, not mRNA level). SRF expression level was also be detected.
9. In the HOPX-overexpressed NPC cells, the author should modulated the SRF expression and observed the changes of Snail and its -mediated EMT, including EMT function and molecular mechanism (protein level, not mRNA level). At the same time, HDCA expression needed to be tested.
10. In Fig.8c, IHC was used to examine HDCA2 and SRF protein expressions
11. There were some syntactic errors in article. Please revised them.

Reviewer #3 (Remarks to the Author)

This manuscript identifies HOPX methylation as a mechanism by which expression of the transcription factor HOPX is lost in nasopharyngeal carcinoma (NPC) and that this epigenetic event correlates with poor outcome in NPC patients. First, the authors demonstrate that HOPX is hypermethylated in tumor tissues from patients with NPC and cell lines. By using gain of function (overexpressing HOPX) in NPC cell lines and loss of function (siRNA against HOPX) in NPEC (non-malignant) cells, the authors showed that HOPX inhibits NPC malignant migration and invasion in vitro, by suppressing SNAIL expression and epithelial to mesenchymal transition (EMT). Repression of SNAIL transcription by HOPX is regulated in cooperation with HDAC2. Finally, the authors demonstrate that HOPX re-expression inhibits NPC metastatic colonization of the lungs in vivo.

The molecular pathogenesis of NPC is poorly understood and thus the scope of the study is significant. Although the role of HOPX as a tumor suppressor has been described in other cancers, its mechanism of action is also poorly understood. Overall this manuscript presents a logical mechanism by which loss of HOPX regulates the activation of SNAIL, a known EMT regulating transcription factor in NPC. Other strengths of the study include extensive supportive data from human biospecimens and a comprehensive set of molecular experiments. However major weakness in the manuscript remain, particularly with regards to the approach used and interpretation of major biological experiments addressing the control of metastasis and EMT by HOPX. Indeed several seminal studies recently proved that EMT is not required for metastatic colonization (Fischer KR, *Nature*, 2015; Zheng X, *Nature*, 2015; Beerling E, *Cell Reports*, 2016). This issue has not been adequately addressed in this manuscript given the state of the field and hypothesis set forth by its authors.

Specific comments that should be addressed are as follows.

Major comments:

1. The biological data used to forward a link between HOPX, EMT, and metastasis are weak. Importantly, the lung metastatic colonization assay used in figure 8 mainly measures extravasation and outgrowth in the lung and does not necessarily reflect invasion or activation of EMT. Therefore, while the fact that HOPX suppresses lung colonization is supportive of its role as a metastasis suppressor, this result also suggest that the mechanism by which it does so may be independent of EMT, or at least also involve other pathways/genes. The readout of invasiveness and EMT would be more accurately and comprehensively supported by additional in vivo assays. Examples include: an orthotopic assay (and measuring entry of NPC cells into circulation from a primary or subcutaneous tumor) and response of tumors to chemotherapy (a feature more closely linked to cancer stem-like phenotypes and EMT).
2. Restoration of HOPX by overexpression is not physiological. The authors should include an in vivo metastatic assay using HOPX loss of function in the context of a non-aggressive cell line (NPEC or NP69, N2Tert or Bmi1).
3. The authors should be more precise in their narrative. There are major papers tackling the role of EMT in cancer and metastasis that should be cited and discussed (see above). Some statements in the introduction and discussion are confusing, and are not consistent with recent literature. Specific examples of overstatements with either inadequate supportive data or inaccurate references include:
 - "To our knowledge, this study is the first investigation of the mechanism of HOPX in the suppression of EMT-mediated cancer metastasis".
 - "However, appropriately 30% of NPC patients with the same TNM stage failed with the similar treatment within 5 years";
 - "Cancer is characterized by a subset of aberrantly expressed genes. Many more latent oncogenes or TSGs upstream or downstream of these TFs exist than TFs themselves".

4. The authors seem to have been biased towards EMT as a biological readout of HOPX function. A more unbiased profiling of HOPX regulated genes may reveal novel and potentially significant mechanisms by which HOPX can suppress metastasis. This is all the more relevant given the fact that EMT is not required for metastasis in many cancers, but rather may promote therapeutic resistance of tumor cells after they have colonized distant organs such as the lungs (see references noted above).

Minor questions/comments:

1. Please include citations of epigenetic silencing of HOPX in malignant tissues or metastasis.
2. Figure: 1 Representation of Bisulfite sequencing data could be improved using a visual diagram of the analyzed CpG islands in HOPX promoter. Also, annotation of the analyzed region in the HOPX promoter will help the reader.
3. Does methylation of HOPX correlate with high levels of SNAIL in the patient samples?
4. In the legend of Figure 1D, the authors refer to a p-value. However p-value annotation is not represented on the plot.
5. Figure 2, 4, 5, and 7: Most of the GAPDH western blots are saturated. It would be better to present a lower exposure.
6. Figure 4A: The change of morphology in cells with overexpression or siRNA against HOPX is not clear in the pictures presented.
7. Figure 5i: The Co-IP between HOPX and HDAC2 is not particularly convincing with the level of background in the IgG lane. In general, the un-edited versions of the Co-IPs actually seem more interpretable.
8. The authors should include the number of biological replicates in all their figures.
9. Figure 8: The model is not supported by the data: namely that HOPX solely suppresses metastasis via SNAIL and the EMT program.
10. In the discussion the authors should relate and cite the literature on SNAIL and EZH2 regulating E-Cadherin (i.e. Tong ZT et al., *Oncogene* 2012) and previous observations of SNAIL expression and metastasis (i.e. Luo WR et al., *Annals of Surgical Oncology*, 2012).
11. There are several errors in grammar and syntax throughout the manuscript, particularly in the Discussion section.

Response to Reviewers' comments

Reviewer #1 (Remarks to the Author):

The manuscript by Ren et al describes the molecular mechanisms dictating the transformation and metastasis of nasopharyngeal carcinoma. Global analyses of DNA methylation defects led to identification of HOPX transcription factor among the most highly methylated TFs. Authors then follow the HOPX downstream mechanisms, which may influence NPC outcomes. SNAIL was found as a primary target gene, where HOPX cooperates with HDAC2 and SRF to repress it, leading to inhibition of expression of EMT markers and reduced cell migration and invasion phenotypes. In vivo lung tumour colonisation experiments demonstrate lower number of metastatic nodes appearing from cells overexpressing HOPX.

HOPX tumour suppressor function was demonstrated before in other cancers. The role in epithelial to mesenchymal transition (EMT) was suggested as well. Thus the main novelty of the paper comes from documenting HOPX involvement in NPCs and the EMT via direct regulation of SNAIL. The manuscript convincingly demonstrated that the cellular phenotypes produced by reduced expression of HOPX are likely to be caused by activation of SNAIL; however it failed to convince that the HOPX directly regulates SNAIL. The details are discussed below:

1. The main weakness of the argument is the lack of unbiased techniques to support the main mechanism. For example, the molecular complex repressing transcription of SNAIL (HOPX, HDAC2, and SRF) is identified by co-IPs of few selected candidates. This finding could be substantially stronger if unbiased IP-MS approach was used and the HDAC2 and SRF were found among the strongest interactors. Moreover, it is unclear if HOPX regulates other genes in addition to SNAIL, which could impact the phenotype of NPC. This could be addressed if the effects of HOPX overexpression measured by microarray or HOPX ChIP-seq were performed in cells, which naturally express HOPX. By selected examination of few factors it is difficult to build more complete picture of the roles of HOPX in NPCs.

Response:

We thank for the reviewer's comments and appreciate for the helpful suggestions. As the reviewer suggested, a mass spectrometry assay and tumor metastasis PCR array were used to strengthen our findings.

To explore the mechanism of *SNAIL* transcription repression mediated by HOPX, immunoprecipitation plus mass spectrometry was performed in SUNE1 cells with HOPX stably overexpressed. Numerous proteins were found to be HOPX-interacting proteins, among which, HDAC2 was the only member of the HDACs family that might be responsible for the transcriptional repression effects of HOPX (**Table 1**). Thus, TSA treatment, real time RT-PCR, Co-IP, ChIP-real time PCR, western blotting and Transwell assays were performed to confirm that HOPX inhibited *SNAIL* expression via interaction with HDAC2 and the recruitment of histone deacetylase activity. In addition, because HOPX functions as a corepressor; it must regulate gene expression through interactions with other TFs. By examining the promoter region of *SNAIL* bound to HOPX, we identified a serum response element (*SRE*) that was the DNA-binding motif of SRF. Furthermore, SRF was also identified as a HOPX-interacting protein in the mass spectrometry assay (**Table 1**). These findings raised the possibility that SRF, as a transcription factor, might recruit HOPX to the *SNAIL* promoter. To test this hypothesis, we applied real time RT-PCR, luciferase reporter assays, Co-IP, ChIP-real time PCR, Transwell invasion assays, western blotting and TSA treatment assays to confirm that HOPX inhibited NPC EMT and invasiveness via the HDAC-mediated transcriptional repression of SRF-dependent *SNAIL* transcription.

Table 1: The HOPX-interacting proteins identified using mass spectrometry.

(See the excel document for supplementary Table 2)

Anti-IgG and anti-HOPX antibodies were performed for immunoprecipitations. Anti-IgG antibody was considered a control. Genes were ranked by scores.

In this study, we found that HOPX could suppress NPC cell migration and invasion and had a limited effect on cell growth *in vitro*. Notably, when HOPX expression was knocked down in NPEC and SH cells, the morphology of some cells transitioned from an epithelial-like form to a spindle-shaped or elongated, mesenchymal form, which indicated that HOPX functioned to maintain the epithelial status of the NPEC and NPC cells. Therefore, we speculated that EMT might be involved in the suppression effect of HOPX on NPC cell invasiveness. To explore the possible downstream effectors that HOPX modulated to suppress the invasiveness of NPC cells, a tumor metastasis PCR array (Qiagen, Germany) was used. Among the top 10 differentially expressed genes (**Table 2**), *ECADHERIN* was among the most significantly upregulated cell adhesion genes in SUNE1 cells with HOPX stably overexpressed (**Fig. 1**), which suggested that HOPX might repress the EMT of NPC cells. Furthermore, both immunofluorescent staining and western blotting assays verified the repressive effects of HOPX on EMT. Considering the critical roles of EMT-TFs in the process of triggering EMT and metastasis, to further determine the target genes of HOPX, we examined the levels of the known EMT-TF regulators (*SNAIL*, *SLUG*, *ZEB1*, *ZEB2*, *TWIST1* and *FOXC2*) after altering HOPX expression. Notably, among all EMT regulators, *SNAIL* was the only regulator exhibiting significant changes following HOPX transfection in all investigated cell lines. Next, western blotting, real time RT-PCR, Co-IP, luciferase reporter assays, ChIP-PCR, ChIP-real time PCR, wound healing assays, and Transwell assays were performed to confirm that HOPX suppressed the invasiveness and EMT of NPC cells via the inhibition of *SNAIL* transcription.

Table 2: The top 10 differentially expressed genes in SUNE1 cells with HOPX stably overexpressed

Gene	Functional Gene Grouping	Fold change	P value
CXCL12	Cytokines	3.26595684	0.000110047
IL1B	Cell Cycle Regulation; Negative Regulation of Cell Proliferation; Cytokines; Apoptosis	48.6101703	0.000908618
NME4	Other Genes Related to Metastasis	37.14376425	0.001387345
MMP2	Matrix Metalloproteinases	21.40828154	0.001792716
EPHB2	Receptors	3.133358068	0.002630822
SERPINE1	Other ECM Proteins	10.77413353	0.003052426
COL4A2	Other ECM Proteins	4.537641016	0.004445524
ECADHERIN	Cell to Cell Adhesion	2.213765777	0.004685254
SYK	Cell to Cell Adhesion; Other Genes Related to Growth	0.265441999	0.005936684
CD44	Cell to Cell Adhesion; Transmembrane Receptors	3.385831498	0.006076176

Data are presented as the mean \pm SD; P < 0.05 compared with the control using Student's *t*-tests; genes were ranked by P value.

Fig. 1: HOPX upregulates ECADHERIN expression in NPC cells. The mRNA level of ECADHERIN identified using a tumor metastasis PCR array in SUNE1 cells with stably overexpressed the vector or HOPX. Data are presented as the mean \pm SD. *, $P < 0.01$ compared with the control using Student's *t*-tests. These data are representative of three independent experiments.

Moreover, many HOPX-interacting proteins and downstream genes of HOPX were identified in the mass spectrometry assay and tumor metastasis PCR array. In this study, we demonstrated that the HOPX repressive effects on NPC invasiveness and EMT depended strongly on the recruitment of the histone deacetylase HDAC2 and the enhancement of histone H3K9 deacetylation of SRF-dependent *SNAIL* transcription. However, other molecular events might also be involved in NPC invasiveness and metastasis, which we will investigate further in future work.

Based on the reviewer's comments, we performed the tumor metastasis PCR array and mass spectrometry to support our main mechanism, and the relevant results have been added to our revised manuscript (Page 8, Paragraph 1; Page 9, Paragraph 3; Page 10, Paragraph 2; Page 17, Paragraph 2; Page 19, Paragraph 1; Supplementary Table 1 and 2; Supplementary Figure 6b; Supplementary methods).

2. It is surprising that around 20% elevated DNA methylation between the cell lines correlates with nearly 150 fold decrease in expression. Since only 20% alleles in the population of cells are methylated, even if methylated allele would silence the gene completely, it is difficult to explain such a dramatic reduction in expression. This implies that other silencing or defective activation mechanism is responsible for the downregulation of HOPX expression.

Response:

Thanks very much for the reviewer's valuable comments. We regret the error in the annotation of the Y-axis in Fig. 2d, and we have changed 100, 150 and 200 to 50, 100 and 150 on the Y-axis in our revised manuscript.

There are two potential explanations. First, *HOPX* was identified as the most significantly altered TF based on the genome-wide methylation microarray analysis in NPC tissues, and only the most significantly altered CpG site (cg21899596) was selected for further methylation analysis in the NPEC and NPC cell lines. Thus, the methylation status of other *HOPX* CpG sites might affect the *HOPX* expression in the NPC cell lines. Second, only 20% methylated alleles could not completely explain such a dramatic reduction in *HOPX* expression completely, and it is possible that some other mechanisms may also be responsible for the downregulation of *HOPX* in NPC, such as gene mutations, deletions, miRNAs, or post-transcriptional regulation, which will be further examined in our future research.

3. The direct interactions between HOPX and HDAC2, SRF seems very weak if any (considering the strength of IgG signals). Endogenous co-IPs from HOPX expressing cell lines (NP69 and N2-Tert) should

be produced to convince that the interactions are real.

Response:

Thanks for the thoughtful suggestions. As the reviewer suggested, we have optimized and repeated the Co-IP assay to verify that HOPX could interact strongly with HDAC2 and SRF in CNE2 and SUNE1 cells with HOPX exogenous expression (CH and SH cells). We also conducted a Co-IP assay to examine the interaction between HOPX and HDAC2 or SRF in the NPEC cells with HOPX endogenous expression (N2-Tert and NP69 cells). The results showed that HOPX could physically interact with HDAC2 and SRF in both NPC and NPEC cells (**Fig. 2**).

To address the reviewer's concerns, we have added the above results to our revised manuscript (Page 10, paragraph 1; Page 11, paragraph 2; Fig. 5g; Fig. 6c; Supplementary Fig. 8a).

Fig. 2: HOPX physically interacts with HDAC2 and SRF in NPC and NPEC cells. Co-IP assay were used to test the interaction between HOPX and HDAC2 in CH, SH, N2-Tert and NP69 cells. CH and SH cells indicated CNE2 and SUNE1 cells with HOPX exogenous expression. Anti-HOPX antibody was used to pull down HOPX; anti-IgG antibody was used as a control. Western blotting (WB) was conducted to examine (a) HOPX, (b) HDAC2 and (c) SRF.

4. Fig 1d, Fig 2f, g. Methylation of HOPX in cell lines should be shown as actual values instead of relative to NP69. The actual values are important for interpretation of expression data in fig 2d. Since experiments were done using pyrosequencing, the values should be readily available.

Response:

Thanks for the reviewer's valuable comments. The primary bisulfite pyrosequencing results for the HOPX CpG site methylation in all cell lines treated with DAC or not are shown in Supplementary Fig. 1b and 2a. Thus, as the reviewer's suggests, we have used the actual values of the HOPX methylation levels in cell lines instead of the values relative to NP69 in Fig. 1e of our revised manuscript. However, to better address the relationships of HOPX methylation and mRNA expression change in NPEC and NPC cells after treatment with DAC, we have still used the relative methylation level and expression level of the HOPX instead of actual values in Fig. 2f and g.

5. Fig 5f ChIP experiments should be quantified with quantitative PCR techniques

Response:

We appreciate the recommendation. We optimized the experimental conditions and re-conducted ChIP-PCR, and we identified only one promoter region (-521 to -680bp) of *SNAIL* as interacting strongly with HOPX (**Fig. 3a-b**). In addition, as the reviewer suggests, we performed ChIP-quantitative PCR to quantify the binding efficiencies between HOPX and the *SNAIL* promoter region (-521 to -680bp), and found that HOPX could physically bind to the promoter region of *SNAIL* (**Fig. 3c**).

To address the reviewer's concern, we have added the above results to our revised manuscript (Page 9, Paragraph 2; Fig. 5f; Supplementary Fig. 7c-e; Supplementary methods).

Fig. 3: HOPX physically binds to the promoter region of *SNAIL* in NPC and NPEC cells. (a) ChIP assay using an anti-HOPX antibody was performed to pull down HOPX. Western blotting (WB) was conducted to examine HOPX using an anti-HOPX Antibody. **(b-c)** ChIP-PCR assay was conducted to assess the enrichment of HOPX in different promoter regions of *SNAIL* in CH, SH, N2-Tert and NP69 cells. **(d)** ChIP- real time PCR assays were conducted to quantify the enrichment of HOPX in the *SNAIL* promoter region. CH and SH cells indicated CNE2 and SUNE1 cells with HOPX exogenous expression.

Data are presented as the mean \pm SD. *, $P < 0.01$ compared with the control using Student's t-tests. These data are representative of three independent experiments.

Reviewer #2 (Remarks to the Author):

In this manuscript, Ren et al observe that promoter hypermethylation lead to the inactivation of HOPX in NPC. Restoration of HOPX inhibits NPC cell migration, invasion, and metastasis through recruiting HDAC2 suppressing SNAIL-mediated EMT. Finally, they analyze HOPX promoter hypermethylation is correlated with clinical features and poor prognosis of NPC patients. Their results demonstrate the significance of HOPX in NPC. Unfortunately, the biggest problem for their paper is that their core data markedly lack of original innovation for mechanism study in key points.

Key Comments:

1. HOPX as a tumor suppressor suppressing cell proliferation, migration, and invasion has been reported in some tumors including pancreatic cancer, uterine endometrial cancer, and gastric cancer, colorectal cancer, lung cancer et al., which is attributed to its promoter hypermethylation. Thus, promoter hypermethylation of HOPX in NPC is not a novel finding.

Response:

Thanks very much for the reviewer's valuable comments. HOPX has been reported to be downregulated due to its promoter hypermethylation and to act as a tumor suppressor suppressing cell proliferation, migration, and invasion in several cancers. However, HOPX has also been reported to be upregulated in thyroid cancer (Pauws E, et al. Thyroid. 2004), sarcoma (Kovarova D, et al. Mol Cancer Res. 2013) and invasive pancreatic cancer (Walsh N, et al. Cancer Lett. 2011), and to increase tumor cell migration and invasion. Thus, the roles of HOPX in different cancer types remain controversial and warrant further study. More importantly, the mechanism of HOPX in cancer is poorly understood.

There are three novel findings in our present study. 1) This study is the first to investigate the different methylated transcriptional factors (TFs) between NPC and normal tissues based on genome-wide methylation microarray analysis, in which HOPX was identified as the most significantly altered TF. HOPX hypermethylation resulted in its downregulation, and the ectopic expression of HOPX suppressed NPC cell migration and invasion with no obvious effect on cell proliferation, suggesting that HOPX acts as a metastasis suppressor in NPC; 2) we demonstrated, for the first time, that the repressive effects of HOPX on NPC invasiveness and EMT depend on the recruitment of the histone deacetylase HDAC2 and the enhancement of histone H3K9 deacetylation of SRF-dependent *SNAIL* transcription; and 3) this report is the first to indicate that HOPX hypermethylation was a prognostic biomarker for NPC patients with a high risk of metastasis, enrolling a large sample size from two hospitals.

2. In this study, the observation about "HOPX epigenetically suppresses SNAIL transcription via HDAC2 recruitment" should be the most core result for this paper. The authors found HOPX epigenetically suppresses SNAIL transcription via recruiting HDAC2, but the interactions for HOPX with HDAC2 (Cardiac hypertrophy and histone deacetylase-dependent transcriptional repression mediated by the atypical homeodomain protein Hop. *J Clin Invest.* 2003; 112(6): 863-71; Hopx and Hdac2 interact to modulate Gata4 acetylation and embryonic cardiac myocytes proliferation. *Dev Cell.* 2010; 19(3): 450-9.), as well as HDAC2 with SNAIL (Repression of 15-hydroxyprostaglandin dehydrogenase involves histone deacetylase 2 and snail in colorectal cancer. *Cancer Res.* 2008; 68(22):9331-7; E-cadherin regulates metastasis of pancreatic cancer in vivo and is suppressed by a SNAIL/HDAC1/HDAC2 repressor complex. *Gastroenterology*, 2009; 137:361-71; Snail mediates E-cadherin repression by the recruitment of the Sin3A/histone deacetylase 1 (HDAC1)/HDAC2 complex. *Mol Cell Biol.* 2004; 24:306-19; Hepatitis C virus core protein interacts with Snail and histone deacetylases to promote the metastasis of hepatocellular carcinoma. *Oncogene*, 2015, Nov 9. [Epub ahead of print]; EZH2 supports nasopharyngeal carcinoma cell aggressiveness by forming a co-repressor complex with HDAC1/HDAC2 and Snail to inhibit E-cadherin. *Oncogene.* 2012; 31(5):583-94; HDAC2 provides a critical support to malignant progression of hepatocellular carcinoma through feedback control of mTORC1 and AKT. *Cancer Res.* 2014; 74(6): 1728-38; A novel domain in histone deacetylase 1 and 2 mediates repression of cartilage-specific genes in human chondrocytes. *FASEB J.* 2009; 23:3539-52) had been reported. Furthermore, the interaction of HDCA2 with SRF was also reported (Homeobox gene HOPX is epigenetically silenced in human uterine endometrial cancer and suppresses estrogen-stimulated proliferation of cancer cells by inhibiting serum response factor. *Int J Cancer.* 2009; 124(11):2577-88.). Finally, the research about "SRF modulated Snail via binding to its promoter" had also been documented (Serum response factor accelerates the high glucose-induced Epithelial-to-Mesenchymal Transition (EMT) via snail signaling in human peritoneal mesothelial cells. *PLoS One*, 2014; 9:e108593.)

Response:

Thanks for the reviewer's valuable comments. We agree that some of the epistatic regulation relationship have been identified. However, the relationship between HOPX and SNAIL remains unknown. Furthermore, despite substantial evidences indicating that HOPX is associated with multiple cancers, the mechanism of HOPX in cancer remains elusive. NPC is characterized by a high rate of local invasion and early distant metastasis. However, the underlying mechanism governing NPC metastasis remains unclear. In this study, for the first time, we demonstrated a novel signaling cascade involved in NPC metastasis, thus HOPX epigenetically inhibit SRF-dependent SNAIL transcription via the recruitment of HDAC2 and the enhancement of histone H3K9 deacetylation. Furthermore, the regulation of HOPX to SNAIL expression is validated in clinical specimens, which provides a potential therapeutic target for NPC. Therefore, the HOPX mediated signaling pathway enriches our understanding of HOPX and NPC metastasis.

Other comments:

1. Suppressing HOPX in the human immortalized normal nasopharyngeal epithelial cell (NPEC) lines induces to a spindle-shaped or elongated, mesenchymal form, why NPC cells with reduced or absent HOPX expression do not indicate this change?

Response:

Thanks for the reviewer's valuable comments. In this study, when the expression of HOPX was knocked down in N2-Tert, NP69 and SH cells, we found that the morphology of some cells transitioned

from an epithelial-like form to a spindle-shaped or elongated, mesenchymal form, which indicated that HOPX functioned to maintain an epithelial-like form of NPC cells. Many genes have been reported to be aberrantly expressed in NPC cell lines compared with human immortalized normal nasopharyngeal epithelial cell lines, among which some could repress EMT, such as YPEL3 (Zhang J, et al. J Exp Clin Cancer Res. 2016), mir-374a (Zhen Y, et al. Oncogene. 2016) and PSCA (Wang L, et al. J Pathol. 2015), while some could promote EMT, such as c-SRC (Ke L, et al. Oncotarget. 2016), BMI-1 (Song L B, et al. J Clin Invest. 2009), EZH2 (Tong Z T, et al. Oncogene. 2012), and COX-2 (Li Z L, et al. Oncoimmunology. 2015). Thus, the effect of HOPX on NPC cell morphology might be compensated by genes that could promote EMT, with the final morphology of NPC cells determined by the effects of multiple genes. Moreover, in our previous study (Song L B, et al. J Clin Invest. 2009), we found that the primary NPEC cells had to be transfected with BMI-1 several times to acquire a typical EMT morphological change, although the molecular markers had clearly changed, which indicated that the typical EMT morphological change was difficult to induce in NPEC and NPC cells. Collectively, these phenomena may be part of the reason NPC cells with reduced HOPX expression do not adopt a typical EMT morphology.

2. NPC is an EBV-related tumor. LMP1 is a critical EBV oncogenic gene in NPC. Whether LMP1 modulated HOPX expression? If yes, How about the mechanism?

Response:

As the reviewer suggested, we transfected CNE2 and SUNE1 cells with LMP1 expression plasmid or vector control and then determined the expression levels of HOPX using real time RT-PCR and western blotting assays. The results showed that LMP1 could not affect HOPX expression at neither the mRNA nor protein levels (**Fig. 1**).

Fig. 1: LMP1 has little effect on HOPX expression. LMP1 or vector expression plasmids were transiently transfected into CNE2 and SUNE1 cells. (a) Real time RT-PCR and (b) western blotting assays were used to test the relative mRNA and protein levels of HOPX and LMP1 in CNE2 and SUNE1 cells. Data are presented as the mean \pm SD. *, $P < 0.01$ compared with the control using Student's t-tests. These data are representative of three independent experiments.

3. In fig. 3 and fig. 4, HOPX expression levels did not be tested in knockdown HOPX cells. Why you did not examine the expression change?

Response:

Thanks for the careful observation. The HOPX expression levels were evaluated after HOPX was overexpressed or silenced in all cell lines (**Fig. 2**). We regret having failed to show our data clearly and completely. The silencing efficiencies of HOPX in N2-Tert and NP69 cells are shown in Fig. 4d, while the restoration efficiencies of HOPX in CNE2 and SUNE1 cells are shown in Fig. 4e of our primary manuscript. In addition, as the reviewer suggested, we have also added the expression changes of HOPX in HONE1, SH and CH cells to our revised manuscript (Supplementary Fig. 3a and Fig 4a).

Fig. 2: The restoring and silencing efficiencies of HOPX in all cell lines. CH and SH cells indicated CNE2 and SUNE1 cells with HOPX exogenous expression.

4. In fig. 5f for CNE2 ChIP, no obvious difference was observed for snail DNA level in IgG and HOPX lanes. Thus, EMSA should be used to confirm the combination of HOPX with Snail via recruiting HDAC2.

Response:

Thanks for the kind suggestions. We regret that the difference in the *SNAIL* DNA level in IgG and HOPX lanes for CNE2 ChIP in Fig.5f appears less than obvious because of the non-specific bands in the IgG group, which may attenuate the persuasiveness of our findings. Thus, we optimized the experimental conditions and re-conducted ChIP-PCR (Fig. 3a-c), and we found that only one promoter region (-521 to -680bp) of *SNAIL* was interacted strongly with HOPX (Fig. 3b). Furthermore, we conducted a ChIP-PCR assay to examine the interaction between HOPX and *SNAIL* promoter in NPEC cells with HOPX endogenous expression (Fig. 3c). Moreover, ChIP-real time PCR was used to quantify the binding efficiency between HOPX and *SNAIL* (Fig. 3d). Our findings demonstrated that HOPX could obviously bind to the promoter of *SNAIL*.

To address the reviewer’s concern, we have added the above results to our revised manuscript (Page 9, Paragraph 2; Fig. 5f; Supplementary Fig. 7c-e; Supplementary methods).

Fig. 3: HOPX physically binds to the promoter region of *SNAIL* in NPC and NPEC cells. (a) ChIP assay using an anti-HOPX antibody was performed to pull down HOPX. Western blotting (WB) was conducted to examine HOPX using an anti-HOPX Antibody. (b-c) ChIP-PCR assay was conducted to assess the enrichment of HOPX in different promoter regions of *SNAIL* in CH, SH, N2-Tert and NP69 cells. (d) ChIP- real time PCR assays were conducted to quantify the enrichment of HOPX in the *SNAIL* promoter region. CH and SH cells indicated CNE2 and SUNE1 cells with HOPX exogenous expression.

Data are presented as the mean \pm SD. *, P < 0.01 compared with the control using Student’s t-tests. These data are representative of three independent experiments.

5. Although the authors confirm the interaction of HOPX with HDAC2 in NPC, they did not confirm whether HOPX modulated the expression of HDCA2? If HOPX modulated the expression of HDCA2, please clarify the molecular basis.

Response:

Thanks for the reviewer’s thoughtful suggestion. As recommended, western blotting assay was used to detect the effects of HOPX on HDAC2 and SRF expression, which were shown to be minimal (Fig. 4).

To address the reviewer’s concern, we have added the above results to our revised manuscript (Page 10, Paragraph 1; Page 11, Paragraph 2; Supplementary Fig. 8b; Supplementary Fig. 9c).

Fig. 4: HOPX has little effect on HDAC2 and SRF expression.

6. The author pointed out that HOPX interacted with SRF by endogenous CoIP. However, GST-PULL Down was needed to further confirm the interactions between HOPX with SRF.

Response:

Thanks for the reviewer’s valuable comments. In this study, we performed IP followed by mass spectrometry (**Table 1**) and Co-IP to investigate the interactions between HOPX and SRF in NPC cells. Furthermore, we verified this interaction in NPEC cells with HOPX endogenous expression (**Fig. 5**). Moreover, the physical interactions between HOPX and SRF in vitro had been well studied using a GST-pull down assay previously (Chen F, et al. Cell. 2002; Shin C H, et al. Cell. 2002); thus, in this study, we did not perform a further GST-pull down assay. We wish to inquire whether the reviewer considers this decision acceptable. If the reviewer insists on a new GST-pull down assay to confirm the interactions between HOPX and SRF, we will perform the experiment.

To address the reviewer’s concerns, we have added the above results to our revised manuscript (Page 8, paragraph 1; Page 9, paragraph 3; Page 10, paragraph 1-2; Page 11, paragraph 2; Page 19, paragraph 1; Fig. 5g; Fig. 6c; Supplementary Table 2; Supplementary Fig. 8a).

Table 1: The HOPX-interacting proteins identified using mass spectrometry.

(See the excel document for supplementary Table 2)

Anti-IgG and anti-HOPX antibodies were performed for immunoprecipitations. Anti-IgG antibody was considered an endogenous control. Genes were ranked by scores.

Fig. 2: HOPX physically interacts with HDAC2 and SRF in NPC and NPEC cells. Co-IP assay were used to test the interaction between HOPX and HDAC2 in CH, SH, N2-Tert and NP69 cells. CH and SH cells indicated CNE2 and SUNE1 cells with HOPX exogenous expression. Anti-HOPX antibody was used to pull down HOPX; anti-IgG antibody was used as a control. Western blotting (WB) was conducted to examine (a) HOPX, (b) HDAC2 and (c) SRF.

7. Endogenous immuno-co-location should be performed among HOPX, HDCA2, and SRF.

Response:

As the reviewer suggested, we performed immunofluorescent staining and found co-location of HOPX, HDAC2 and SRF in NPC cells with stable HOPX overexpression (**Fig. 6**).

To address the reviewer’s concerns, the above results have been added to our revised manuscript (Page 12, Paragraph 2; Supplementary Fig. 10).

Fig. 6: HOPX is co-localized with HDAC2 and SRF in NPC cells. Immunofluorescence images (600×) for HOPX (red), HDAC2 (purple) and SRF (green) expression in CNE2 and SUNE1 cells stably overexpressed vector or HOPX. These data are representative of three independent experiments.

8. *In the HOPX-overexpressed NPC cells, the author should modulate the HDCA2 expression and observe the changes of Snail and its-mediated EMT, including EMT function and molecular mechanism (protein level, not mRNA level). SRF expression level was also be detected.*

Response:

Thanks for the considerate comments. As the reviewer suggested, we transfected small interfering RNA for HDAC2 (siHDAC2) or siNC into the HOPX-overexpression NPC cells (CH and SH), and we found that co-transfection with siHDAC2 could significantly abolish the inhibitory effects on invasion in HOPX-overexpression NPC cells (**Fig. 7a**). In addition, the expression of SNAIL suppressed by HOPX was substantially increased, while the expression of ECADHERIN was decreased following co-transfection with siHDAC2. However, there was little effect of HDAC2 on SRF expression (**Fig. 7b**). Therefore, these

findings illustrate that HDAC2 is an important functional molecule in HOPX modulation of NPC invasiveness and EMT.

To address the reviewer's concerns, we have added the relevant result to our revised manuscript (Page 10, Paragraph 1; Supplementary Fig. 8c,d).

Fig. 7: Silencing HDAC2 could reverse the inhibition ability of HOPX on invasiveness and EMT in NPC. (a) Transwell assay with Matrigel (200 \times) were used to measure the invasion abilities of CH and SH cells which were transfected with siHDAC2 or siNC. **(b)** Western blotting was used to examine the expression levels of SNAIL, ECADHERIN, HDAC2, SRF, HOPX and GAPDH in CH and SH cells which were transfected with siHDAC2 or siNC. CH and SH cells indicated CNE2 and SUNE1 cells with HOPX overexpression. Data are presented as the mean \pm SD. *, $P < 0.01$ compared with the control using the Student's t-tests. These data are representative of three independent experiments.

9. In the HOPX-overexpressed NPC cells, the author should modulate the SRF expression and observe the changes of Snail and its-mediated EMT, including EMT function and molecular mechanism (protein level, not mRNA level). At the same time, HDCA expression needed to be tested.

Response:

Thanks for the considerate comments. As the reviewer suggested, we transfected SRF expression plasmid or vector control into the HOPX-overexpression NPC cells (CH and SH) and found that the restoring SRF expression could significantly enhance cell invasion and EMT (Fig. 8a-b). However, SRF exhibited little effect on HDAC2 expression (Fig. 8b). Collectively, these findings demonstrate that SRF is an important functional molecule in HOPX modulation of NPC invasiveness and EMT.

To address the reviewer's concerns, we have added the relevant result to our revised manuscript (Page 12, Paragraph 1; Supplementary Fig. 9f,g).

Fig. 8: Restoring SRF could reverse the inhibition ability of HOPX on invasiveness and EMT in NPC. (a, b) Transwell assay with Matrigel (200 \times) were used to measure the invasion abilities of CH and SH cells which were transfected with SRF expression plasmid or vector control. **(c, d)** Western blotting was used to examine the expression levels of SNAIL, ECADHERIN, HDAC2, SRF, HOPX and GAPDH in CH and SH cells which were transfected with SRF expression plasmid or vector control. CH and SH cells indicated CNE2 and SUNE1 cells with HOPX overexpression. Data are presented as the mean \pm SD. *, $P < 0.01$ compared with the control using Student's t-tests. These data are representative of three independent

experiments.

10. In Fig.8c, IHC was used to examine HDCA2 and SRF protein expressions.

Response:

Thanks for the valuable comments. To address the reviewer's concerns, IHC was used to assess HDAC2 and SRF protein levels in the lung sections. The results demonstrated that HDAC2 and SRF protein levels showed little difference in the HOPX-overexpression group compared with the control group (**Fig. 9**).

Fig. 9: HOPX has little effect on HDAC2 and SRF expression.

11. There were some syntactic errors in article. Please revise them.

Response:

Thanks for the reviewer's valuable comments. Prior to resubmission, the manuscript was proofread by two scientific editors who work for a professional language editing service (NPG editing service, <https://languageediting.nature.com/>). We have marked the revised portions in red in our revised manuscript.

Reviewer #3 (Remarks to the Author):

This manuscript identifies HOPX methylation as a mechanism by which expression of the transcription factor HOPX is lost in nasopharyngeal carcinoma and that this epigenetic event correlates with poor outcome in NPC patients. First, the authors demonstrate that HOPX is hypermethylated in tumor tissues from patients with NPC and cell lines. By using gain of function (overexpressing HOPX) in NPC cell lines and loss of function (siRNA against HOPX) in NPEC (non-malignant) cells, the authors showed that HOPX inhibits NPC malignant migration and invasion in vitro, by suppressing SNAIL expression and epithelial to mesenchymal transition (EMT). Repression of SNAIL transcription by HOPX is regulated in cooperation with HDAC2. Finally, the authors demonstrate that HOPX re-expression inhibits NPC metastatic colonization of the lungs in vivo.

The molecular pathogenesis of NPC is poorly understood and thus the scope of the study is significant. Although the role of HOPX as a tumor suppressor has been described in other cancers, its mechanism of action is also poorly understood. Overall this manuscript presents a logical mechanism by which loss of HOPX regulates the activation of SNAIL, a known EMT regulating transcription factor in NPC. Other strengths of the study include extensive supportive data from human biospecimens and a comprehensive set of molecular experiments. However major weakness in the manuscript remains, particularly with regards to the approach used and interpretation of major biological experiments addressing the control of metastasis and EMT by HOPX. Indeed several seminal studies recently proved that EMT is not required for metastatic

colonization (Fischer KR, Nature, 2015; Zheng X, Nature, 2015; Beerling E, Cell Reports, 2016). This issue has not been adequately addressed in this manuscript given the state of the field and hypothesis set forth by its authors.

Response:

Thanks very much for the reviewer's valuable comments. Before the submission of this manuscript, we did notice these two papers. We also extended our knowledge regarding EMT in cancer metastasis through several recently published articles.

The involvement of EMT in cancer metastasis has been debated for a long time, mainly because of the difficulty of tracking its process *in vivo*. Recently, researchers from two centers provided provocative results opposing the importance of EMT in metastasis (Fischer K R, et al. Nature. 2015; Zheng X, et al. Nature. 2015). However, EMT could not yet be ruled out as a driver of cancer progression for the following reasons (Nieto M A, et al. Cell. 2016; Li W, et al. Trends Cancer. 2016; Maheswaran S, et al. Nature. 2015). 1) the definition of EMT has been broadened to include some spectrum of intermediary phases during this transition, which have been suggested to involve more aggressiveness and easier metastatic dissemination compared with the complete epithelial or mesenchymal states (Grigore A D, et al. J Clin Med. 2016; Huang R Y, et al. Cell Death Dis. 2013). Furthermore, tracing cells based on a single factor may not completely catch all ongoing EMT events, especially those intermediate EMT phenotypes; 2) a recent study from the Beerling E. group demonstrates that some breast tumor cells that spontaneously undergo EMT acquire metastatic motility, then subsequently disseminate and revert back to the epithelial state upon metastatic colonization (Beerling E, et al. Cell Rep. 2016); 3) EMT may still be required for tumor progression to the metastatic state because it could facilitate the invasion of other tumor cells (Nieto M A, et al. Cell. 2016); 4) EMT can be regulated by a comprehensive network, such as EMT-TFs, noncoding RNAs, epigenetic regulators and other post transcriptional molecules, which indicates that the loss of individual factors may be unable to prevent cell invasion and dissemination (Maheswaran S, et al. Nature. 2015); 5) mesenchymal markers are upregulated in CTCs, demonstrating that the invasion and intravasation abilities of those cells were enhanced (Nieto M A, et al. Cell. 2016); and 6) cancer is characterized as a highly heterogeneous disease; the degree of partial EMT required for metastatic dissemination by different cancer types may be not the same (Li W, et al. Trends Cancer. 2016). Therefore, additional studies are needed to verify the contribution of EMT as a driver of cancer progression.

In this study, we demonstrate that the repression on EMT and invasiveness by HOPX may depend on the recruitment of the histone deacetylase HDAC2 and the enhancement of histone H3K9 deacetylation of SRF-dependent *SNAIL* transcription. *SNAIL* is among the most important EMT-TFs and has been demonstrated to exhibit a substantially broader effect on embryonic development and tumor progression (Wu Y, et al. Cell Adh Migr. 2010; Wang Y, et al. Curr Cancer Drug Targets. 2013). Not surprisingly, *SNAIL* also has a vital effect on NPC progression. High nuclear *SNAIL* expression has been reported to be significantly associated with poor survival in NPC (Luo W R, et al. Ann Surg Oncol. 2012). Numerous regulators exert their functions on metastasis through regulating *SNAIL* in NPC, such as BMI-1 (Song L B, et al. J Clin Invest. 2009), EZH2 (Tong Z T, et al. Oncogene. 2012), LMP1 (Horikawa T, et al. Br J Cancer. 2011), EBV-mir-BART10-3p (Yan Q, et al. Oncotarget. 2015), EBV-mir-BART7-3p (Cai L M, et al. Oncogene. 2015), CACNA2D3 (Wong A M, et al. Int J Cancer. 2013) and ME1 (Zheng F J, et al. Chin J Cancer. 2012). *BMI-1* has been reported to induce EMT via stabilizing *SNAIL* through the PI3K-AKT-GSK3 β signaling pathway in NPC cells, which ultimately enhanced the lung metastatic colonization ability of the tumor cells (Song L B, et al. J Clin Invest. 2009). EZH2 could form a complex with HDAC1, HDAC2 and *SNAIL* to enhance NPC cell metastasis *in vitro* and lung metastatic colonization

in vivo (Tong Z T, et al. Oncogene. 2012). Therefore, a comprehensive understanding of the regulatory mechanism of *SNAIL* expression will provide vital information regarding EMT blockage and the suppression of metastasis in NPC.

Because the roles of EMT in cancer metastasis remain unclear, we have corrected our imprecise description of “EMT-mediated metastasis” or “metastasis via EMT”, etc., throughout the paper, and we have added some citations of recent literature to strengthen and support our hypothesis. This topic is further discussed in our revised manuscript (Title; Abstract; Page 5, Paragraph 1-2; Page 7, Paragraph 3; Page 8, Paragraph 1-3; Page 9, Paragraph 2; Page 12, Paragraph 1; Page 16, Paragraph 1; Page 17, Paragraph 2; Page 18, Paragraph 1-2; Page 20, Paragraph 2).

Specific comments that should be addressed are as follows:

Major comments:

1. The biological data used to forward a link between HOPX, EMT, and metastasis is weak. Importantly, the lung metastatic colonization assay used in figure 8 mainly measures extravasation and outgrowth in the lung and does not necessarily reflect invasion or activation of EMT. Therefore, while the fact that HOPX suppresses lung colonization is supportive of its role as a metastasis suppressor, this result also suggest that the mechanism by which it does so may be independent of EMT, or at least also involve other pathways/genes.

*The readout of invasiveness and EMT would be more accurately and comprehensively supported by additional *in vivo* assays. Examples include: an orthotopic assay (and measuring entry of NPC cells into circulation from a primary or subcutaneous tumor) and response of tumors to chemotherapy (a feature more closely linked to cancer stem-like phenotypes and EMT).*

Response:

Thanks for the reviewer’s valuable comments. We agree that the biological data used to support a link between HOPX, EMT, and metastasis are weak, and the results for invasiveness and EMT would be more accurately and comprehensively supported by additional *in vivo* assays. Because there is no exact nasopharyngeal tissue in mice, we used an inguinal lymph node metastasis model to imitate the orthotopic model (Cao S, et al. Oncotarget. 2016). The inguinal lymph node metastasis model was constructed through injecting SUNE1 cells that stably overexpressed the vector or HOPX into the foot pads of mice. After 6-weeks of growth, the primary foot pad tumors and inguinal lymph nodes were obtained (n = 8 / group, **Fig. 1a**). H&E staining of the primary tumors showed that the tumors in the HOPX-overexpression group exhibited sharp edges that expanded as spheroids, which indicated a less aggressive phenotype with invasion towards the skin, muscle and lymphatic vessel than the vector group (**Fig. 1b** and **Fig. 2**). Furthermore, the inguinal lymph nodes in the HOPX-overexpression group had smaller volumes (**Fig. 1c**) and lower numbers of pan-cytokeratin-positive tumor cells (**Fig. 1d**) compared with the vector group. Additionally, the inguinal lymph node metastasis ratio is significantly lower in the HOPX-overexpression group (**Fig. 1e**). Taken together, these findings demonstrate that HOPX suppresses NPC cell invasion and lymph node metastasis *in vivo*.

Fig. 1: HOPX restoration inhibits NPC cell aggressiveness and SNAIL expression *in vivo*. SUNE1 cells with vector or HOPX overexpression (n = 8 / group) were injected into the foot pads of mice to construct an inguinal lymph node metastasis model. Six weeks later, the foot pad tumors and inguinal lymph nodes were excised for analysis. **(a)** Representative images of the primary foot pad tumor and the metastatic inguinal lymph node. **(b)** Representative images (200×) of the microscopic primary tumor in foot pad stained with hematoxylin and eosin (H&E) staining. **(c)** Representative images and quantification of the average volumes of the inguinal lymph nodes. **(d)** Immunohistochemical staining for pan-cytokeratin-positive tumor cells in the inguinal lymph nodes (40× and 200×). **(e)** Metastatic ratios of inguinal lymph nodes; Chi-square test was used for statistical analysis. Data are presented as the mean ± SD. *, P < 0.01 compared with the control using Student's t-tests.

Fig. 2: HOPX restoration inhibits NPC cell aggressiveness *in vivo*. Representative images (200×) of the microscopic primary tumor in foot pad stained with hematoxylin and eosin (H&E) staining. Arrows represent lymphatic vessels.

In addition, as the reviewer suggested, to determine whether HOPX has any effect on the response of tumors to chemotherapy, we first examined the cisplatin (DDP) sensitivity of NPC cells stably overexpressed the vector or HOPX using MTT assays, and found that restoring HOPX significantly reduced the IC values of DDP, indicating that HOPX could enhance the sensitivity of DDP in NPC cells *in vitro* (**Fig. 3a**). Next, an *in vivo* xenograft tumor model was constructed by injecting CNE2 cells stably overexpressing vector or HOPX into the dorsal flank of the mice. After growing for 3 days, the tumor nodes became palpable (approximately 100 mm³). Then, the mice were randomly divided into 4 groups (n = 5 / group) and injected intraperitoneally with normal saline or DDP every 3 days: Vector + Normal saline; HOPX + Normal saline; Vector + DDP (4 mg/kg); HOPX + DDP (4 mg/kg). Tumor size was measured every 3 days for approximately 2 weeks. Then, the mice were sacrificed, and the tumors were dissected and weighted. In the normal saline-treatment groups, we found that HOPX exhibited no significant effect on tumor growth (**Fig. 3b,c**). However, in the DDP-treatment groups, the tumor volumes (**Fig. 3d**) and weight (**Fig. 3e**) were significantly suppressed in the HOPX-overexpressing group compared with the vector group. Together,

these findings imply that HOPX could enhance the cisplatin sensitivity of NPC cells.

Fig. 3: HOPX enhances the sensitivity of NPC cells to cisplatin treatment. (a) MTT assays were used to test the effects of DDP on CNE2 and SUNE1 cell with vector or HOPX stable overexpression. (b) Representative picture of xenograft tumors in nude mice treated with normal saline or DDP (4mg/kg). (c-d) The growth curves of the tumor volumes in (c) normal saline treatment group and (d) DDP treatment group. (e) The tumor weight in normal saline treatment group and DDP treatment group. Data are presented as the mean \pm SD. *P < 0.01 compared with control using the Student *t*-test. These data are representative of three independent experiments.

To address the reviewer's concern, we have added the relevant results to our revised manuscript (Page 12, Paragraph 4; Page 13, Paragraph 1; Page 22, Paragraph 1; Fig. 8a-e; Supplementary Fig. 11).

2. Restoration of HOPX by overexpression is not physiological. The authors should include an *in vivo* metastatic assay using HOPX loss of function in the context of a non-aggressive cell line (NPEC or NP69, N2Tert or Bmi1).

Response:

Thanks for the reviewer's valuable comments. We agree that the restoration of HOPX by overexpression is not physiological, and we should construct an *in vivo* metastatic assay using HOPX loss of function in the context of a non-aggressive cell line (NPEC or NP69, N2Tert or Bmi1). However, we regret that we could not finish this experiment because the non-aggressive cell line (NPEC or NP69, N2Tert or Bmi1) could not colonize and metastasize to distant tissues *in vivo*.

3. The authors should be more precise in their narrative. There are major papers tackling the role of EMT in cancer and metastasis that should be cited and discussed (see above). Some statements in the introduction and discussion are confusing, and are not consistent with recent literature. Specific examples of overstatements with either inadequate supportive data or inaccurate references include:

- "To our knowledge, this study is the first investigation of the mechanism of HOPX in the suppression of EMT-mediated cancer metastasis".

- "However, appropriately 30% of NPC patients with the same TNM stage failed with the similar treatment within 5 years";

- "Cancer is characterized by a subset of aberrantly expressed genes. Many more latent oncogenes or TSGs upstream or downstream of these TFs exist than TFs themselves".

Response:

Thanks for the reviewer's helpful comments. As mentioned above, because the role of EMT in cancer metastasis is still unclear, we have corrected our imprecise description of "EMT-mediated metastasis" or "metastasis via EMT", etc., throughout the paper, and we have added some recent literature citations to strengthen and support our hypothesis. This topic is further discussed in our revised manuscript. We have also corrected the confusing statements in the introduction and discussion of our revised manuscript. (Title; Abstract; Page 4, Paragraph 1-2; Page 5, Paragraph 1-2; Page 7, Paragraph 3; Page 8, Paragraph 1-3; Page 9, Paragraph 2; Page 16, Paragraph 1-3; Page 17, Paragraph 1-2; Page 18, Paragraph 1-2; Page 20, Paragraph 2).

4. The authors seem to have been biased towards EMT as a biological readout of HOPX function. A more unbiased profiling of HOPX regulated genes may reveal novel and potentially significant mechanisms by which HOPX can suppress metastasis. This is all the more relevant given the fact that EMT is not required for metastasis in many cancers, but rather may promote therapeutic resistance of tumor cells after they have colonized distant organs such as the lungs (see references noted above).

Response:

Thanks for the reviewer's helpful suggestions. As the reviewer suggested, a tumor metastasis PCR array and mass spectrometry were used to strengthen our findings.

First, in this study, we found that HOPX could suppress NPC cell migration and invasion and had a limited effect on cell growth *in vitro*. Notably, when HOPX expression was knocked down in NPEC and SH cells, the morphology of some cells transitioned from an epithelial-like form to a spindle-shaped or elongated, mesenchymal form, which indicated that HOPX functioned to maintain the epithelial status of the NPEC and NPC cells. Therefore, we speculated that EMT might be involved in the suppression effect of HOPX on NPC cell invasiveness. To explore the possible downstream effectors that HOPX modulated to suppress the invasiveness of NPC cells, a tumor metastasis PCR array containing 84 metastasis-associated genes (Qiagen, Germany) was used. Among the top 10 differentially expressed genes (**Table 1**), *ECADHERIN* was among the most significantly upregulated cell adhesion genes in SUNE1 cells with HOPX stably overexpressed (**Fig.4**), which suggested that HOPX might repress the EMT of NPC cells. Furthermore, both immunofluorescent staining and western blotting assays verified the repressive effects of HOPX on EMT. Considering the critical roles of EMT-TFs in the process of triggering EMT and metastasis, to further determine the target genes of HOPX, we examined the levels of the known EMT-TF regulators (*SNAIL*, *SLUG*, *ZEB1*, *ZEB2*, *TWIST1* and *FOXC2*) after altering HOPX expression. Notably, among all EMT regulators, *SNAIL* was the only regulator exhibiting significant changes following HOPX transfection in all investigated cell lines. Next, western blotting, real time RT-PCR, Co-IP, luciferase reporter assays, ChIP-PCR, ChIP-real time PCR, wound healing assays, and Transwell assays were performed to confirm that HOPX suppressed the invasiveness and EMT of NPC cells via the inhibition of *SNAIL* transcription.

Table 1: The top 10 differentially expressed genes in SUNE1 cells with HOPX stably overexpressed

Gene	Functional Gene Grouping	Fold change	P value
CXCL12	Cytokines	3.26595684	0.000110047
IL1B	Cell Cycle Regulation; Negative Regulation of Cell Proliferation; Cytokines; Apoptosis	48.6101703	0.000908618
NME4	Other Genes Related to Metastasis	37.14376425	0.001387345
MMP2	Matrix Metalloproteinases	21.40828154	0.001792716
EPHB2	Receptors	3.133358068	0.002630822
SERPINE1	Other ECM Proteins	10.77413353	0.003052426
COL4A2	Other ECM Proteins	4.537641016	0.004445524
ECADHERIN	Cell to Cell Adhesion	2.213765777	0.004685254
SYK	Cell to Cell Adhesion; Other Genes Related to Growth	0.265441999	0.005936684
CD44	Cell to Cell Adhesion; Transmembrane Receptors	3.385831498	0.006076176

Data are presented as the mean \pm SD; $P < 0.05$ compared with the control using Student's *t*-tests; genes were ranked by P value.

Fig. 4: HOPX upregulates ECADHERIN expression in NPC cells. The mRNA level of *ECADHERIN* identified using a tumor metastasis PCR array in SUNE1 cells with vector or HOPX stably overexpressed. Data are presented as the mean \pm SD. *, $P < 0.01$ compared with the control using Student's *t*-tests. These data are representative of three independent experiments.

Second, to explore the mechanism of *SNAIL* transcription repression mediated by HOPX, immunoprecipitation plus mass spectrometry was performed in SUNE1 cells with HOPX stably overexpressed. Numerous proteins were found to be HOPX-interacting proteins, among which, HDAC2 was the only member of the HDACs family that might be responsible for the transcriptional repression effects of HOPX (**Table 2**). Thus, TSA treatment, real time RT-PCR, Co-IP, ChIP-real time PCR, western blotting and Transwell assays were performed to confirm that HOPX inhibited *SNAIL* expression via interaction with HDAC2 and the recruitment of histone deacetylase activity. In addition, because HOPX functions as a corepressor; it must regulate gene expression through interactions with other TFs. By examining the promoter region of *SNAIL* bound to HOPX, we identified a serum response element (*SRE*) that was the DNA-binding motif of SRF. Furthermore, SRF was also identified as a HOPX-interacting protein in the mass spectrometry assay (**Table 2**). These findings raised the possibility that SRF, as a transcription factor, might recruit HOPX to the *SNAIL* promoter. To test this hypothesis, we applied real time RT-PCR, luciferase reporter assays, Co-IP, ChIP-real time PCR, Transwell invasion assays, western blotting and TSA treatment

assays to confirm that HOPX inhibited NPC EMT and invasiveness via the HDAC-mediated transcriptional repression of SRF-dependent *SNAIL* transcription.

Table 2: The HOPX-interacting proteins identified using mass spectrometry

(See the excel document for supplementary Table 2)

Anti-IgG and anti-HOPX antibodies were performed for immunoprecipitations. Anti-IgG antibody was considered a control. Genes were ranked by scores.

Moreover, many downstream genes and interacting proteins of HOPX were identified in the tumor metastasis PCR array and mass spectrometry assays. In this study, we demonstrated that the HOPX repressive effects on NPC invasiveness and EMT depended strongly on the recruitment of the histone deacetylase HDAC2 and the enhancement of histone H3K9 deacetylation of SRF-dependent *SNAIL* transcription. However, other molecular events might also be involved in NPC invasiveness and metastasis, which we will investigate further in future work.

Based on the reviewer's comments, we performed the tumor metastasis PCR array and mass spectrometry to support our main mechanism, and the relevant results have been added to our revised manuscript (Page 8, Paragraph 1; Page 9, Paragraph 3; Page 10, Paragraph 2; Page 17, Paragraph 2; Page 19, Paragraph 1; Supplementary Table 1 and 2; Supplementary Figure 6b; Supplementary methods).

Minor questions/comments:

1. Please include citations of epigenetic silencing of HOPX in malignant tissues or metastasis.

Response:

As the reviewer suggested, we have added citations of the epigenetic silencing of HOPX in malignant tissues or metastasis in our revised manuscript (Page 16, Paragraph 3; References 48, 22, 50 and 20).

2. Figure: 1 Representation of Bisulfite sequencing data could be improved using a visual diagram of the analyzed CpG islands in HOPX promoter. Also, annotation of the analyzed region in the HOPX promoter will help the reader.

Response:

Thanks for the reviewer's thoughtful suggestions. We have added a visual diagram of the analyzed CpG islands in the HOPX promoter (**Fig. 5**) in our revised manuscript (Page 6, Paragraph 2; Fig. 1b).

Fig. 5: Schematic of the CpG islands and bisulfite pyrosequencing region in the promoter of HOPX. Input sequence: red region; CpG islands: blue region; TSS: transcription start site; cg21899596, cg24852548 and cg10800833 were the sites of HOPX identified in our previous genome-wide methylation microarrays; red words: CG sites for bisulfite pyrosequencing;

bold red words: the most significantly altered CG site in HOPX.

3. Does methylation of HOPX correlates with high levels of SNAIL in the patient samples?

Response:

Thanks for the valuable comments. To address the reviewer's concern, the methylation levels of HOPX were detected in the same tissues used to determine the HOPX and SNAIL mRNA levels. Pearson correlation analysis indicated that the HOPX methylation level positively correlated with SNAIL expression (Fig. 6). We have added these results to our revised manuscript (Page 14, Paragraph 1; Supplementary Fig. 12).

Fig. 6: The methylation rate of HOPX is positively associated with SNAIL mRNA expression in NPC. (a) Relative HOPX methylation rates were determined via bisulfite sequencing, while SNAIL expression values were determined via real time RT-PCR in the NPC tissues (n = 24). Statistical analysis was performed using the Pearson's coefficient test. Data are presented as the mean \pm SD. These data are representative of three independent experiments.

4. In the legend of Figure 1D, the authors refer to a p-value. However p-value annotation is not represented on the plot.

Response:

Thanks for the reviewer's valuable suggestion. We have shown the actual values of HOPX methylation levels instead of the values relative to NP69, as Reviewer 1 suggested, and have added the symbol "*" to represent a p-value < 0.05, as this reviewer suggested in Fig. 1e of our revised paper.

5. Figure 2,4, 5, and 7: Most of the GAPDH western blots are saturated. It would be better to present a lower exposure.

Response:

Thanks for the reviewer's considerate suggestion. We have accordingly changed the saturated GAPDH western blots to clear ones in Fig. 2e, 4d-e, 5c-d and 7d of our revised manuscript.

6. Figure 4A: The change of morphology in cells with overexpression or siRNA against HOPX is not clear in the pictures presented.

Response:

Thanks for the reviewer's thoughtful suggestions. We have accordingly changed the unclear pictures to clear ones in Fig. 4a of our revised manuscript.

7. Figure 5i: The Co-IP between HOPX and HDAC2 is not particularly convincing with the level of background in the IgG lane. In general, the un-edited versions of the Co-IPs actually seem more interpretable.

Response:

Thanks for the reviewer's valuable comments. To address the reviewer's concerns, we have optimized and repeated the Co-IP assay to verify that HOPX could strongly interact with HDAC2 and SRF in NPC cells with exogenous HOPX expression. Furthermore, we conducted a Co-IP assay to examine the interaction between HOPX and HDAC2 or SRF in the NPEC cells with endogenous HOPX expression according to the Reviewer 1's suggestion. As the results showed, HOPX could physically interact with HDAC2 and SRF in both NPC and NPEC cells (**Fig. 7**).

To address the reviewer's concerns, we have added the above results to our revised manuscript (Page 10, paragraph 1; Page 11, paragraph 2; Fig. 5g; Fig. 6c; Supplementary Fig. 8a).

Fig. 2: HOPX physically interacts with HDAC2 and SRF in NPC and NPEC cells. Co-IP assay were used to test the interaction between HOPX and HDAC2 in CH, SH, N2-Tert and NP69 cells. CH and SH cells indicated CNE2 and SUNE1 cells with HOPX exogenous expression. Anti-HOPX antibody was used to pull down HOPX; anti-IgG antibody was used as a control. Western blotting (WB) was conducted to examine (a) HOPX, (b) HDAC2 and (c) SRF.

8. The authors should include the number of biological replicates in all their figures.

Response:

Thanks for the reviewer's reminder. All data presented as the mean \pm SD were extracted from no less than three independent experiments. We have added the number of biological replicates in corresponding figures, according to the reviewer's suggestion.

9. Figure 8: The model is not supported by the data: namely that HOPX solely suppresses metastasis via SNAIL and the EMT program.

Response:

Thanks for the reviewer's valuable comments. In this study, we found that HOPX suppresses EMT and metastasis in NPC via the recruitment of histone deacetylase HDAC2 to epigenetically inhibit SRF-dependent SNAIL transcription. In addition, there might be some other target genes of HOPX involved in NPC invasiveness and metastasis. We have generated a more precise model to support our data (**Fig. 8**) to replace the original figure; this new figure is Fig. 10 in our revised manuscript (Page 20, paragraph 2; Fig. 10)

Fig. 8: Schematic summary of the HOPX-HDAC2/SRF-SNAIL signaling pathway. HOPX suppresses EMT and metastasis in NPC via the recruitment of histone deacetylase HDAC2 to epigenetically inhibit SRF-dependent *SNAIL* transcription. This suppression effect could partially release from the *HOPX* hypermethylation.

10. In the discussion the authors should relate and cite the literature on *SNAIL* and *EZH2* regulating *E-Cadherin* (i.e. Tong ZT et al., *Oncogene* 2012) and previous observations of *SNAIL* expression and metastasis (i.e. Luo WR et al., *Annals of Surgical Oncology*, 2012).

Response:

In light of the reviewer's perspective, we have cited and discussed these literatures in the discussion section of our revised manuscript (Page 18, Paragraph 2; Reference 63 and 65).

11. There are several errors in grammar and syntax throughout the manuscript, particularly in the Discussion section.

Response:

Thanks for the reviewer's valuable comments. Prior to resubmission, the manuscript was proofread by two scientific editors who work for a professional language editing service (NPG editing service, <https://languageediting.nature.com/>). We have marked the revised portions in red in our revised manuscript.

Yours faithfully,
Jun Ma and Na Liu

References

- 1 Pauws, E., Sijmons, G. G., Yaka, C. & Ris-Stalpers, C. A novel homeobox gene overexpressed in thyroid carcinoma. *Thyroid*. **14**, 500-505 (2004).
- 2 Kovarova, D. *et al.* Downregulation of HOPX controls metastatic behavior in sarcoma cells and identifies genes associated with metastasis. *Mol Cancer Res*. **11**, 1235-1247 (2013).
- 3 Walsh, N. *et al.* RNAi knockdown of Hop (Hsp70/Hsp90 organising protein) decreases invasion via MMP-2 down regulation. *Cancer Lett*. **306**, 180-189 (2011).
- 4 Zhang, J. *et al.* YPEL3 suppresses epithelial-mesenchymal transition and metastasis of nasopharyngeal carcinoma cells through the Wnt/beta-catenin signaling pathway. *J Exp Clin Cancer Res*. **35**, 109 (2016).
- 5 Zhen, Y. *et al.* miR-374a-CCND1-pPI3K/AKT-c-JUN feedback loop modulated by PDCD4 suppresses cell growth, metastasis, and sensitizes nasopharyngeal carcinoma to cisplatin. *Oncogene*. (2016).
- 6 Wang, L. *et al.* Down-regulation of prostate stem cell antigen (PSCA) by Slug promotes metastasis in nasopharyngeal carcinoma. *J Pathol*. **237**, 411-422 (2015).
- 7 Ke, L. *et al.* c-Src activation promotes nasopharyngeal carcinoma metastasis by inducing the epithelial-mesenchymal transition via PI3K/Akt signaling pathway: a new and promising target for NPC. *Oncotarget*. **7**, 28340-28355 (2016).
- 8 Song, L. B. *et al.* The polycomb group protein Bmi-1 represses the tumor suppressor PTEN and induces epithelial-mesenchymal transition in human nasopharyngeal epithelial cells. *J Clin Invest*. **119**, 3626-3636 (2009).
- 9 Tong, Z. T. *et al.* EZH2 supports nasopharyngeal carcinoma cell aggressiveness by forming a co-repressor complex with HDAC1/HDAC2 and Snail to inhibit E-cadherin. *Oncogene*. **31**, 583-594 (2012).
- 10 Li, Z. L. *et al.* COX-2 promotes metastasis in nasopharyngeal carcinoma by mediating interactions between cancer cells and myeloid-derived suppressor cells. *Oncoimmunology*. **4**, e1044712 (2015).
- 11 Chen, F. *et al.* Hop is an unusual homeobox gene that modulates cardiac development. *Cell*. **110**, 713-723 (2002).
- 12 Shin, C. H. *et al.* Modulation of cardiac growth and development by HOP, an unusual homeodomain protein. *Cell*. **110**, 725-735 (2002).
- 13 Fischer, K. R. *et al.* Epithelial-to-mesenchymal transition is not required for lung metastasis but contributes to chemoresistance. *Nature*. **527**, 472-476 (2015).
- 14 Zheng, X. *et al.* Epithelial-to-mesenchymal transition is dispensable for metastasis but induces chemoresistance in pancreatic cancer. *Nature*. **527**, 525-530 (2015).
- 15 Nieto, M. A., Huang, R. Y., Jackson, R. A. & Thiery, J. P. EMT: 2016. *Cell*. **166**, 21-45 (2016).
- 16 Li, W. & Kang, Y. Probing the Fifty Shades of EMT in Metastasis. *Trends Cancer*. **2**, 65-67 (2016).
- 17 Maheswaran, S. & Haber, D. A. Cell fate: Transition loses its invasive edge. *Nature*. **527**, 452-453 (2015).
- 18 Grigore, A. D., Jolly, M. K., Jia, D., Farach-Carson, M. C. & Levine, H. Tumor Budding: The Name is EMT. Partial EMT. *J Clin Med*. **5** (2016).
- 19 Huang, R. Y. *et al.* An EMT spectrum defines an anoikis-resistant and spheroidogenic intermediate mesenchymal state that is sensitive to e-cadherin restoration by a src-kinase inhibitor, saracatinib (AZD0530). *Cell Death Dis*. **4**, e915 (2013).
- 20 Beerling, E. *et al.* Plasticity between Epithelial and Mesenchymal States Unlinks EMT from

- Metastasis-Enhancing Stem Cell Capacity. *Cell Rep.* **14**, 2281-2288 (2016).
- 21 Wu, Y. & Zhou, B. P. Snail: More than EMT. *Cell Adh Migr.* **4**, 199-203 (2010).
- 22 Wang, Y., Shi, J., Chai, K., Ying, X. & Zhou, B. P. The Role of Snail in EMT and Tumorigenesis. *Curr Cancer Drug Targets.* **13**, 963-972 (2013).
- 23 Luo, W. R., Li, S. Y., Cai, L. M. & Yao, K. T. High expression of nuclear Snail, but not cytoplasmic staining, predicts poor survival in nasopharyngeal carcinoma. *Ann Surg Oncol.* **19**, 2971-2979 (2012).
- 24 Horikawa, T. *et al.* Epstein-Barr Virus latent membrane protein 1 induces Snail and epithelial-mesenchymal transition in metastatic nasopharyngeal carcinoma. *Br J Cancer.* **104**, 1160-1167 (2011).
- 25 Yan, Q. *et al.* EBV-miR-BART10-3p facilitates epithelial-mesenchymal transition and promotes metastasis of nasopharyngeal carcinoma by targeting BTRC. *Oncotarget.* **6**, 41766-41782 (2015).
- 26 Cai, L. M. *et al.* EBV-miR-BART7-3p promotes the EMT and metastasis of nasopharyngeal carcinoma cells by suppressing the tumor suppressor PTEN. *Oncogene.* **34**, 2156-2166 (2015).
- 27 Wong, A. M. *et al.* Characterization of CACNA2D3 as a putative tumor suppressor gene in the development and progression of nasopharyngeal carcinoma. *Int J Cancer.* **133**, 2284-2295 (2013).
- 28 Zheng, F. J. *et al.* Repressing malic enzyme 1 redirects glucose metabolism, unbalances the redox state, and attenuates migratory and invasive abilities in nasopharyngeal carcinoma cell lines. *Chin J Cancer.* **31**, 519-531 (2012).
- 29 Cao, S. *et al.* Upregulation of flotillin-1 promotes invasion and metastasis by activating TGF-beta signaling in nasopharyngeal carcinoma. *Oncotarget.* **7**, 4252-4264 (2016).

Reviewer #1 (Remarks to the Author)

Authors have thoroughly addressed the comments. New supplementary figures 1 and 2 provide absolute measurements of DNA methylation in the cell lines examined. I still think it is necessary to have absolute numbers in main figures 2 f, g both for DNA methylation and mRNA levels. Otherwise reader can be left confused by discrepancy of Western blot in panel e demonstrating high protein levels in NP69, N2-Tert and N2-Bmi1 cell lines and normalised values of DNA methylation and mRNA in f and g.

Reviewer #2 (Remarks to the Author)

In revised manuscript, Ren et al supplemented some data and thus improved the quality of this manuscript to a certain extent. However, I still insist on my opinion that the authors did not present a very novel core finding although the authors declaimed that HOPX modulated SNAIL through a novel signaling cascade "HOPX epigenetically inhibit SRF-dependent SNAIL transcription via the recruitment of HDAC2 and the enhancement of histone H3K9 deacetylation". In the previous review, I had found that the interaction for HOPX with HDAC2, the interaction of HDCA2 with SRF, and SRF modulated Snail via binding to its promoter had been reported in some documents and the authors just connected them together and formed a signaling cascade "HOPX/HDAC2/SRF/SNAIL according to these documents. Thus, this manuscript for the novelty in signaling cascade "HOPX/HDAC2/SRF/SNAIL" was weak and was not good enough to allow for publication on Nature communications.

Reviewer #3 (Remarks to the Author)

The authors addressed most of the reviewer's comments and the manuscript has been significantly improved. A few remaining issues related to the new data should be addressed prior to accepting the manuscript for publication:

- 1) The authors do not report if the differences in LN invasion are due to differences in engraftment/and or size of tumors at the foot pad. Despite the significant differences in local and LN invasion, the data is confounding if the tumor area/volume at the site of injection is also affected by HOPX.
- 2) The authors performed a qPCR array to evaluate broader differences in tumor metastasis related genes upon Hopx overexpression. Among these genes, CDH1 was increased in cells expressing high levels of Hopx as predicted by their model. However, the authors should comment on the apparent HOPX mediated activation of other genes, such as CD44, CXCL12 and IL1B, which paradoxically are known to promote invasion and tumorigenesis.
- 3) From the rebuttal letter, Figure 3 entitled "HOPX enhances the sensitivity of NPC cells to cisplatin treatment" is an important set of findings that should be reported in the manuscript. Describing the effects of HOPX on chemosensitivity significantly increases the clinical and biological impact of the study and is consistent with one of several biological consequences of EMT and tumor metastasis.
- 4) Almost all the figure legends include the following generic statement regarding sampling: "Data are presented as the mean \pm SD. These data are representative of three independent experiments", even in figures were this statement is unlikely to have been accurate. For example Suppl Fig 12 is presumably a single analysis of patient samples, with no mean and SD. Please revise your each figure legends for accuracy.

Response to Reviewers' comments

Reviewer #1 (Remarks to the Author)

Authors have thoroughly addressed the comments. New supplementary figures 1 and 2 provide absolute measurements of DNA methylation in the cell lines examined. I still think it is necessary to have absolute numbers in main figures 2 f, g both for DNA methylation and mRNA levels. Otherwise reader can be left confused by discrepancy of Western blot in panel e demonstrating high protein levels in NP69, N2-Tert and N2-Bmi1 cell lines and normalised values of DNA methylation and mRNA in f and g.

Response:

We thank for the reviewer's valuable comments and appreciate for the helpful suggestions. As the reviewer suggested, we have exhibited the actual values of the *HOPX* methylation levels and mRNA levels in **Fig. 1**, and the relevant results have been added to our revised manuscript (Figure 2f,g).

Figure 1. (a,b) *HOPX* methylation levels measured via bisulfite pyrosequencing analysis (a) and relative *HOPX* mRNA levels measured via real time RT-PCR analysis (b) with (DAC+) or without (DAC-) DAC treatment in human immortalized NPEC and NPC cell lines. NPEC cell lines: NP69, N2-Tert and N2-Bmi1; NPC cell lines: SUNE1, CNE1, CNE2, HNE1 and HONE1. Mean \pm s.e.m.; *, $P < 0.01$ compared with the DAC-; Student's *t*-tests. These data are representative of three independent experiments.

Reviewer #2 (Remarks to the Author)

In revised manuscript, Ren et al supplemented some data and thus improved the quality of this manuscript to a certain extent. However, I still insist on my opinion that the authors did not present a very novel core finding although the authors declaimed that *HOPX* modulated *SNAIL* through a novel signaling cascade “*HOPX* epigenetically inhibit *SRF*-dependent *SNAIL* transcription via the recruitment of *HDAC2* and the enhancement of histone *H3K9* deacetylation”. In the previous review, I had found that the interaction for *HOPX* with *HDAC2*, the interaction of *HDAC2* with *SRF*, and *SRF* modulated *Snail* via binding to its promoter had been reported in some documents and the authors just connected them together and formed a signaling cascade “*HOPX/HDAC2/SRF/SNAIL*” according to these documents. Thus, this manuscript for the novelty in signaling cascade “*HOPX/HDAC2/SRF/SNAIL*” was weak and was not good enough to allow for publication on *Nature communications*.

Response:

Thanks for the reviewer's valuable comments. Except some epistatic regulation relationship have been identified previously as the reviewer listed, there are three novel findings in this manuscript. 1) This study is the first to indicate that *HOPX* suppresses metastasis and enhances chemosensitivity of NPC cell; 2) this study is the first to demonstrate that the repression effect of *HOPX* may depend on epigenetically inhibiting *SNAIL* transcription; and 3) this report is the first to indicate that *HOPX* hypermethylation is a prognostic biomarker for NPC patients enrolling a large sample size from two hospitals.

NPC is characterized by a high rate of local invasion and early distant metastasis, and the treatment outcomes of metastatic patients remain frustrating. However, the molecular events enrolling in NPC metastasis remain unclear. EMT, which can be triggered by EMT-TFs, has been commonly thought to be a crucial process for metastasis and chemoresistance in cancer. *SNAIL* is one of the most important EMT-TFs and has been demonstrated to exhibit a vital effect on NPC progression. However, the regulation networks of *SNAIL* in NPC remain elusive. In this study, for the first time, we raise and verify the critical role of *HOPX* on *HOPX*/*HDAC2*/*SRF*/*SNAIL* signaling cascade with comprehensive and detailed data, which enriches our understanding of *HOPX* on NPC metastasis and chemosensitivity, as well as the regulation network of *SNAIL*. We believe the scope of this study is significant and will help us establish new insight into NPC progression.

Reviewer #3 (Remarks to the Author)

The authors addressed most of the reviewer's comments and the manuscript has been significantly improved. A few remaining issues related to the new data should be addressed prior to accepting the manuscript for publication:

*1) The authors do not report if the differences in LN invasion are due to differences in engraftment/and or size of tumors at the foot pad. Despite the significant differences in local and LN invasion, the data is confounding if the tumor area/volume at the site of injection is also affected by *HOPX*.*

Response:

Thanks for the reviewer's thoughtful comments. When performing the inguinal lymph node metastasis model, we examined the volumes of the primary foot pad tumors and found that there was no significant difference between *HOPX* overexpression group and the vector group (**Fig. 1, $P > 0.05$**). We have added the relevant result to our revised manuscript to address the reviewer's concern (Page 12, Paragraph 2; Supplementary Fig. 11a).

Figure 1. Restoring *HOPX* expression has little effect on the growth of the primary foot pad tumors compared with the vector group *in vivo*. Quantification of the average volumes of the primary foot pad tumors. Mean \pm s.d.; $P > 0.05$; Student's *t*-tests.

2) *The authors performed a qPCR array to evaluate broader differences in tumor metastasis related genes upon Hopx overexpression. Among these genes, CDH1 was increased in cells expressing high levels of Hopx as predicted by their model. However, the authors should comment on the apparent HOPX mediated activation of other genes, such as CD44, CXCL12 and IL1B, which paradoxically are known to promote invasion and tumorigenesis.*

Response:

Thanks for the reviewer's valuable comments. A tumor metastasis PCR array showed that *ECADHERIN* was upregulated in NPC cells with stable HOPX overexpression. In addition, the expressions of some other tumor suppressors (*NME4* and *CD82*) and oncogenes (*SYK*, *CD44*, *CXCL12* and *IL1B*) were also changed by HOPX. For example, HOPX significantly reduced the expression of *SYK*. High *SYK* level is associated with poor clinical outcomes of NPC patients (Du Z M, et al. Head Neck. 2012). However, *CD44* was also upregulated by HOPX which could promote NPC progression (Shen Y A, et al. Oncotarget. 2016). We suggested that tumor cells naturally kept balance in expressing the tumor suppressors and oncogenes. Exogenous HOPX overexpression disrupted this homeostasis. While tumor suppressors were strengthened by ectopic HOPX expression, some oncogenes were accordingly activated as a feedback to keep the malignant features of cancer cells by the unknown mechanism. This similar phenomenon could be seen in many other studies (Kovarova D, et al. Mol Cancer Res. 2013; Cheung W K, et al. Cancer Cell. 2013; Lu J, et al. J Clin Invest. 2015). Thus, the effects of HOPX are complicated and are determined by the combined effects of multiple downstream genes on NPC progression. In this study we mainly focused on the effects of HOPX on *SNAIL* expression. However, other molecular events might also be involved in NPC progression, which will be investigated in our future work (Page 17, Paragraph 1).

3) *From the rebuttal letter, Figure 3 entitled "HOPX enhances the sensitivity of NPC cells to cisplatin treatment" is an important set of findings that should be reported in the manuscript. Describing the effects of HOPX on chemosensitivity significantly increases the clinical and biological impact of the study and is consistent with one of several biological consequences of EMT and tumor metastasis.*

Response:

Thanks for the reviewer's thoughtful opinion. As the reviewer suggested, we have added the results of the effects of HOPX on chemosensitivity (**Fig. 2**) to our revised manuscript (Abstract; Page 4, Paragraph 1-2; Page 5, Paragraph 1-2; Page 13, Paragraph 3; Page 14, Paragraph 1; Page 15, Paragraph 3; Page 16, Paragraph 2; Page 17, Paragraph 2; Page 18, Paragraph 1; Page 19, Paragraph 2; Figure 9; Methods; Figure 9).

Figure 2. HOPX enhances the sensitivity of NPC cells to DDP. (a) Dose-response curves of CNE2 and SUNE1 cells that stably overexpressed the vector or HOPX following DDP (0, 0.625, 1.25, 2.5, 5 and 10 $\mu\text{g ml}^{-1}$) treatment. Mean \pm s.d.; *, $P < 0.01$ compared with vector; Student's t -tests. (b-e) CNE2 cells stably overexpressed the vector or HOPX were injected into the dorsal flank of the mice. The mice were randomly divided into 4 groups ($n = 5$ per group) and treated with normal saline or DDP (4 mg kg^{-1}). These data are representative of five independent experiments (each mouse sample was considered as an independent experiment). (b) Representative picture of xenograft tumors in nude mice. (c,d) The growth curves of the tumor volumes in normal saline (c) or DDP (d) treatment group. Mean \pm s.d.; *, $P < 0.01$ compared with vector; Student's t -tests. (e) The tumor weights in normal saline or DDP treatment group. Mean \pm s.d.; *, $P < 0.01$ compared with vector; Student's t -tests. (f) Schematic summary of the HOPX-HDAC2/SRF-SNAIL signaling pathway. HOPX suppresses metastasis and enhances chemosensitivity in NPC via the recruitment of histone deacetylase HDAC2 to epigenetically inhibit SRF-dependent *SNAIL* transcription. This suppression effect could partially release from the *HOPX* hypermethylation.

4) Almost all the figure legends include the following generic statement regarding sampling: "Data are presented as the mean \pm SD. These data are representative of three independent experiments", even in figures where this statement is unlikely to have been accurate. For example Suppl Fig 12 is presumably a single analysis of patient samples, with no mean and SD. Please revise your each figure legends for accuracy.

Response:

Thanks for the reviewer's considerate suggestion. To address the reviewer's concern, we have changed the inaccurate description "Data are presented as the mean \pm SD. These data are representative of three independent experiments" to a more accurate description in all of the figure legends of our revised manuscript (Figure legends and Supplementary Figure legends).

References:

- 1 Du, Z. M. *et al.* Clinical significance of elevated spleen tyrosine kinase expression in nasopharyngeal carcinoma. *Head Neck*. **34**, 1456-1464 (2012).
- 2 Shen, Y. A. *et al.* CD44 and CD24 coordinate the reprogramming of nasopharyngeal carcinoma cells towards a cancer stem cell phenotype through STAT3 activation. *Oncotarget*. (2016).
- 3 Kovarova, D. *et al.* Downregulation of HOPX controls metastatic behavior in sarcoma cells and identifies genes associated with metastasis. *Mol Cancer Res*. **11**, 1235-1247 (2013).
- 4 Cheung, W. K. *et al.* Control of alveolar differentiation by the lineage transcription factors GATA6 and HOPX inhibits lung adenocarcinoma metastasis. *Cancer Cell*. **23**, 725-738 (2013).
- 5 Lu, J. *et al.* IRX1 hypomethylation promotes osteosarcoma metastasis via induction of CXCL14/NF-kappaB signaling. *J Clin Invest*. **125**, 1839-1856 (2015).